# The infection-tolerant white-footed deermouse tempers interferon responses to endotoxin in comparison to the mouse and rat

Ana Milovic[1], Jonathan V Duong[1], Alan G Barbour[2]*

[1]Department of Microbiology & Molecular Genetics, University of California, Irvine, Irvine, United States; [2]Departments of Medicine, Microbiology & Molecular Genetics, and Ecology & Evolutionary Biology, University of California, Irvine, Irvine, United States

*For correspondence:
abarbour@uci.edu

Competing interest: The authors declare that no competing interests exist.

**Abstract** The white-footed deermouse *Peromyscus leucopus*, a long-lived rodent, is a key reservoir in North America for agents of several zoonoses, including Lyme disease, babesiosis, anaplasmosis, and a viral encephalitis. While persistently infected, this deermouse is without apparent disability or diminished fitness. For a model for inflammation elicited by various pathogens, the endotoxin lipopolysaccharide (LPS) was used to compare genome-wide transcription in blood by *P. leucopus*, *Mus musculus,* and *Rattus norvegicus* and adjusted for white cell concentrations. Deermice were distinguished from the mice and rats by LPS response profiles consistent with non-classical monocytes and alternatively-activated macrophages. LPS-treated *P. leucopus*, in contrast to mice and rats, also displayed little transcription of interferon-gamma and lower magnitude fold-changes in type 1 interferon-stimulated genes. These characteristics of *P. leucopus* were also noted in a *Borrelia hermsii* infection model. The phenomenon was associated with comparatively reduced transcription of endogenous retrovirus sequences and cytoplasmic pattern recognition receptors in the deermice. The results reveal a mechanism for infection tolerance in this species and perhaps other animal reservoirs for agents of human disease.

## eLife assessment

This study provides a comprehensive whole genome transcriptomic analysis of three small mammals, including Peromyscus leucopus, after exposure to endotoxin lipopolysaccharide. The authors find that the inflammatory response of the three species is complex and that P. leucopus responds differently compared to mice and rats. The data are **convincing** and constitute an **important** advance in our understanding of inflammatory responses in animals that serve as reservoirs for relevant pathogens.

## Introduction

How does the white-footed deermouse *Peromyscus leucopus* continue to thrive while sustaining infections with disease agents it serves as reservoir for (*Barbour, 2017*) ? The diverse tickborne pathogens (and diseases) for humans include the extracellular bacterium *Borreliella burgdorferi* (Lyme disease), the intracellular bacterium *Anaplasma phagocytophilum* (anaplasmosis), the protozoan *Babesia microti* (babesiosis), and the Powassan flavivirus (viral encephalitis). Most deermice remain persistently infected but display scant inflammation in affected tissues (*Cook and Barbour, 2015*; *Long et al.,*

**eLife digest** Lyme disease is an illness caused by bacteria that spread from infected animals to humans through tick bites. While most people fully recover after a week or two of antibiotic treatments, some will continue to experience debilitating symptoms due, potentially, to the way their immune system responded to the infection.

In North America, the white-footed deermouse is one of the most common hosts of the Lyme disease bacteria. Despite its name, this rodent is more closely related to hamsters than to the mice or rats most often used in laboratory studies. Unlike mice and humans, however, deermice carrying Lyme disease bacteria do not get sick; in fact, most deermice living in a Lyme disease region will acquire the infection during their lifetimes, but it has little apparent effect on population numbers. These animals can also better tolerate infection from other microbes.

To investigate why this is the case, Milovic et al. exposed mice, rats and deermice to a bacterial toxin that triggers inflammation common to encounters with many kinds of microbes. While all species exhibited physical symptoms as a result, blood samples revealed that mice and rats, but not deermice, reacted as if they were infected with viruses as well as bacteria. This was particularly the case for interferons, a group of hormone-like proteins that protect against viruses but can also lead to harmful long-term inflammatory effects. The deermice controlled their interferon responses to the bacterial substance in a way that mice and rats could not.

Milovic et al. also checked which genes each species switched on after exposure to the toxin. This revealed that, unlike deer mice, rats and mice turned on some DNA sequences called endogenous retroviruses, which have no role in fighting infection from bacteria but can lead to harmful persistent inflammation.

These results provide elements to better understand why recovery from Lyme disease may differ between people, with some patients retaining symptoms long after their infection has abated. They could also help to better grasp why other diseases, such as COVID-19, can be followed by fatigue and other symptoms of ongoing inflammation.

---

*2019*; *Moody et al., 1994*), and without apparent consequence for fitness (*Schwanz et al., 2011*; *Voordouw et al., 2015*).

A related question—conceivably with the same answer—is what accounts for the two-to-three fold longer life span for *P. leucopus* than for the house mouse, *Mus musculus* (*Labinskyy et al., 2009*; *Sacher and Hart, 1978*) ? The abundance of *P. leucopus* across much of North America (*Hall, 1979*; *Moscarella et al., 2019*) and its adaptation to a variety of environments, including urban areas and toxic waste sites (*Biser et al., 2004*; *Levengood and Heske, 2008*; *Munshi-South and Kharchenko, 2010*), indicates successful adjustment to changing landscapes and climate. *Peromyscus* species, including the hantavirus reservoir *P. maniculatus* (*Morzunov et al., 1998*), are more closely related to hamsters and voles in family Cricetidae than to mice and rats of family Muridae (*Bradley et al., 2014*).

As a species native to North America, *P. leucopus* is an advantageous alternative to the Eurasian-origin house mouse for study of natural variation in populations that are readily accessible (*Bedford and Hoekstra, 2015*; *Long et al., 2022*). A disadvantage for the study of any *Peromyscus* species is the limited reagents and genetic tools of the sorts that are applied for mouse studies. As an alternative, we study *P. leucopus* with a non-reductionist approach that is comparative in design and agnostic in assumptions (*Balderrama-Gutierrez et al., 2021*). The genome-wide expression comparison for *P. leucopus* is with *M. musculus* and, added here, the brown rat *Rattus norvegicus*. Given the wide range of pathogenic microbes that deermice tolerate, we use the bacterial endotoxin lipopolysaccharide (LPS) as the primary experimental treatment because the inflammation it elicits within a few hours has features common to different kinds of serious infections, not to mention severe burns and critical injuries (*Xiao et al., 2011*).

We previously reported that a few hours after injection of LPS, *P. leucopus* and *M. musculus* had distinguishing profiles of differentially expressed genes (DEG) in the blood, spleen, and liver (*Balderrama-Gutierrez et al., 2021*). In brief, the inflammation phenotype of deermice was consistent with an 'alternatively activated' or M2-type macrophage polarization phenotype instead of the expected 'classically activated' or M1-type polarization phenotype that was observed for *M. musculus*

(*Murray et al., 2014*). The deermice also differed from mice in displaying evidence of greater neutrophil activation and degranulation after LPS exposure. The potentially damaging action from neutrophil proteases and reactive oxygen species appeared to be mitigated in part in *P. leucopus* by proteins like secretory leukocyte peptidase inhibitor, encoded by *Slpi*, and superoxide dismutase 2, encoded by *Sod2*.

Here, we first address whether the heightened transcription of neutrophil-associated genes in *P. leucopus* is attributable to differences in numbers of white cells in the blood. To better match for genetic diversity, we substituted outbred *M. musculus* for the inbred BALB/c mouse of the previous study. We retained the experimental protocol of short-term responses to LPS. This main experiment was supplemented by a study of rats under the similar conditions, by an investigation of a different dose of LPS and duration of exposure in another group of deermice, and by analysis of deermice infected with a bacterium lacking LPS. The focus was on the blood of these animals, not only because the distinctions between species in their transcriptional profiles were nearly as numerous for this specimen as for spleen and liver (*Balderrama-Gutierrez et al., 2021*), but also because for ecological and immunological studies of natural populations of *Peromyscus* species blood is obtainable from captured-released animals without their sacrifice.

The results inform future studies of *Peromyscus* species, not only with respect to microbial infections and innate immunity, but conceivably also determinants of longevity and resilience in the face of other stressors, such as toxic substances in the environment. The findings pertain as well to the phenomenon of infection tolerance broadly documented in other reservoirs for human disease agents, such as betacoronaviruses and bats (*Mandl et al., 2018*). Less directly, the results provide for insights about maladaptive responses among humans to microbes, from systemic inflammatory response syndrome to post-infection fatigue syndromes.

## Results

### LPS experiment and hematology studies

Twenty adult animals each for *P. leucopus* and *M. musculus* and equally divided between sexes received by intraperitoneal injection either purified *E. coli* LPS at a dose of 10 μg per g body mass or saline alone (*Table 1*). Within 2 hr LPS-treated animals of both species displayed piloerection and sickness behavior, that is reduced activity, hunched posture, and huddling. By the experiment's termination at 4 hr, 8 of 10 *M. musculus* treated with LPS had tachypnea, while only one of ten LPS-treated *P. leucopus* displayed this sign of the sepsis state (p=0.005).

Within a given species there was little difference between LPS-treated and control animals in values for erythrocytes. But overall the deermice had lower mean (95% confidence interval) hematocrit at 42 (36-48)%, hemoglobin concentration at 13.8 g/dL (12.1–15.5), and mean corpuscular volume for erythrocytes at 49 fL (47-51) than *M. musculus* with respective values of 56 (51-62)%, 16.1 g/dL (14.6–17.7), and 60 fL (58-62) (p<0.01). These hematology values for adult CD-1 *M. musculus* and LL stock *P. leucopus* in this study were close to what had been reported for these colony populations (*Charles-River, 2012*; *Wiedmeyer et al., 2014*).

In contrast to red blood cells, the mean numbers of white blood cells in the LPS groups in both species were lower than those of control groups (*Figure 1*). Controls had a mean 4.9 (3.5–6.4) x $10^3$ white cells per μl among *M. musculus* and 5.8 (4.2–7.4) x $10^3$ white cells per μl among *P. leucopus* (p=0.41). For the LPS-treated animals the values were 2.1 (1.5–2.7) x $10^3$ for mice and 3.1 (0.9–5.4) x $10^3$ for deermice (p=0.39). However, there was difference between species among LPS-treated animals in the proportions of neutrophils and lymphocytes in the white cell population. The ratios of neutrophils to lymphocytes were 0.25 (0.14–0.45) and 0.20 (0.13–0.31) for control *M. musculus* and *P. leucopus*, respectively (p=0.53). But under the LPS condition. the neutrophil-to-lymphocyte ratio was 0.18 (0.11–0.28) for mice and 0.64 (0.42–0.97) for deermice (p=0.0006). The regression curves for plots of neutrophils and lymphocytes for LPS-treated and control *P. leucopus* and LPS-treated *M. musculus* had similar slopes, but the *y*-intercept was shifted upwards towards a higher ratio of neutrophils to lymphocytes for blood from the LPS group of deermice. Control group mice and deermice and LPS-treated mice had similar percentages (~5%) of monocytes in their blood; the mean monocyte percentage rose to 10% in LPS treated deermice (p=0.12). Eosinophil percentages tended to

**Table 1.** Characteristics and treatments of *Mus musculus* CD-1 and *Peromyscus leucopus* LL stock.

| Animal | Genus | Sex | Age (d) | Mass (g) | Treatment | Tachypnea | Hct* (%) | MCV* | WBC* | Neutrophils | Lymphocytes | Monocytes | Eosinophils | Neutrophils/lymphocytes |
|---|---|---|---|---|---|---|---|---|---|---|---|---|---|---|
| MM19 | *Mus* | female | 149 | 60.4 | control† | no | 71 | 62 | 3800 | 570 | 2926 | 190 | 114 | 0.19 |
| MM21 | *Mus* | female | 149 | 51.4 | control | no | 51 | 59 | 8600 | 826 | 4897 | 177 | 0 | 0.17 |
| MM23 | *Mus* | female | 149 | 30.6 | control | no | 51 | . | 1500 | 135 | 1305 | 60 | 0 | 0.10 |
| MM25 | *Mus* | female | 149 | 42.1 | control | no | 65 | 61 | 7900 | 1106 | 6557 | 237 | 0 | 0.17 |
| MM27 | *Mus* | female | 149 | 39.7 | control | no | 58 | 62 | 3700 | 962 | 2257 | 333 | 148 | 0.43 |
| MM1 | *Mus* | male | 149 | 50.9 | control | no | 67 | 59 | 3300 | 726 | 2376 | 198 | 0 | 0.31 |
| MM17 | *Mus* | male | 149 | 40.8 | control | no | 51 | . | 7000 | 4970 | 2030 | 0 | 0 | 2.45 |
| MM3 | *Mus* | male | 149 | 45.8 | control | no | 58 | 59 | 4300 | 344 | 3655 | 301 | 0 | 0.09 |
| MM5 | *Mus* | male | 149 | 41.2 | control | no | 60 | 57 | 5400 | 810 | 4266 | 324 | 0 | 0.19 |
| MM7 | *Mus* | male | 149 | 44.1 | control | no | 47 | 58 | 3800 | 798 | 2736 | 266 | 0 | 0.29 |
| MM31 | *Mus* | female | 149 | 65.5 | LPS | yes | 53 | 58 | 1900 | 209 | 1539 | 114 | 38 | 0.14 |
| MM33 | *Mus* | female | 149 | 48.4 | LPS | yes | 62 | 62 | 3200 | 256 | 2784 | 96 | 64 | 0.09 |
| MM35 | *Mus* | female | 149 | 40.5 | LPS | yes | 49 | 64 | 2200 | 528 | 1518 | 88 | 66 | 0.35 |
| MM37 | *Mus* | female | 149 | 38.5 | LPS | no | 54 | 65 | 1900 | 152 | 1634 | 57 | 57 | 0.09 |
| MM39 | *Mus* | female | 149 | 40.2 | LPS | yes | 61 | 63 | 600 | 60 | 510 | 12 | 18 | 0.12 |
| MM11 | *Mus* | male | 149 | 57.4 | LPS | yes | 59 | 58 | 2600 | 572 | 1898 | 104 | 0 | 0.30 |
| MM13 | *Mus* | male | 149 | 58.5 | LPS | yes | 66 | 59 | 3700 | 1258 | 2183 | 259 | 0 | 0.58 |
| MM15 | *Mus* | male | 149 | 51.0 | LPS | yes | 33 | 56 | 1800 | 90 | 1602 | 90 | 0 | 0.06 |
| MM29 | *Mus* | male | 149 | 39.1 | LPS | no | 53 | 59 | 1800 | 306 | 1368 | 72 | 54 | 0.22 |
| MM9 | *Mus* | male | 149 | 44.1 | LPS | yes | 58 | . | 1300 | 260 | 858 | 182 | 0 | 0.30 |
| 24841 | *Peromyscus* | female | 162 | 19.9 | control | no | 58 | 48 | 5100 | 204 | 4590 | 102 | 204 | 0.04 |
| 24842 | *Peromyscus* | female | 164 | 18.3 | control | no | 47 | 48 | 3900 | 663 | 3003 | 117 | 117 | 0.22 |

*Table 1 continued on next page*

Table 1 continued

| Animal | Genus | Sex | Age (d) | Mass (g) | Treatment | Tachypnea | Hct* (%) | MCV* | WBC* | Neutrophils | Lymphocytes | Monocytes | Eosinophils | Neutrophils/lymphocytes |
|---|---|---|---|---|---|---|---|---|---|---|---|---|---|---|
| 24843 | Peromyscus | female | 162 | 20.5 | control | no | 45 | 46 | 4200 | 942 | 3150 | 42 | 84 | 0.30 |
| 24845 | Peromyscus | female | 161 | 18.7 | control | no | 49 | 48 | 9100 | 2002 | 6916 | 182 | 0 | 0.29 |
| 24853 | Peromyscus | female | 160 | 22.8 | control | no | 42 | . | 4800 | 1008 | 2880 | 864 | 1 | 0.35 |
| 24852 | Peromyscus | male | 162 | 19.4 | control | no | 44 | 46 | 7100 | 994 | 4970 | 284 | 852 | 0.20 |
| 24861 | Peromyscus | male | 157 | 20.8 | control | no | 28 | 50 | 1300 | 104 | 1053 | 91 | 52 | 0.10 |
| 24863 | Peromyscus | male | 157 | 17.1 | control | no | 26 | 58 | 7200 | 720 | 5904 | 288 | 288 | 0.12 |
| 24869 | Peromyscus | male | 143 | 29.0 | control | no | 48 | 52 | 6000 | 1260 | 4260 | 180 | 240 | 0.30 |
| 24876 | Peromyscus | male | 142 | 16.1 | control | no | 54 | 48 | 9700 | 2716 | 6208 | 194 | 485 | 0.44 |
| 24846 | Peromyscus | female | 162 | 22.7 | LPS | no | 23 | 47 | 1100 | 231 | 718 | 44 | 44 | 0.32 |
| 24847 | Peromyscus | female | 162 | 16.7 | LPS | no | 39 | . | 1500 | 570 | 675 | 180 | 75 | 0.84 |
| 24848 | Peromyscus | female | 166 | 16.2 | LPS | no | 43 | 49 | 2700 | 918 | 1701 | 27 | 54 | 0.54 |
| 24850 | Peromyscus | female | 157 | 19.4 | LPS | no | 46 | . | 3300 | 1551 | 1683 | 66 | 0 | 0.92 |
| 24851 | Peromyscus | female | 161 | 25.4 | LPS | no | 24 | 47 | 2300 | 552 | 1242 | 437 | 69 | 0.44 |
| 24855 | Peromyscus | male | 165 | 27.1 | LPS | no | 51 | 51 | 2100 | 1281 | 714 | 42 | 63 | 1.79 |
| 24860 | Peromyscus | male | 160 | 17.8 | LPS | yes | 42 | 54 | 13,200 | 6996 | 3696 | 1320 | 1056 | 1.89 |
| 24865 | Peromyscus | male | 160 | 16.7 | LPS | no | 49 | 46 | 1800 | 396 | 1080 | 288 | 0 | 0.37 |
| 24873 | Peromyscus | male | 145 | 23.2 | LPS | no | 43 | 48 | 2200 | 550 | 1430 | 110 | 110 | 0.38 |
| 24879 | Peromyscus | male | 145 | 22.0 | LPS | no | 40 | . | 1100 | 220 | 538 | 352 | 0 | 0.41 |

*Abbreviations: Hct, hematocrit; MCV, mean cellular volume of erythrocytes; WBC, white blood cell count.

†control, saline alone.

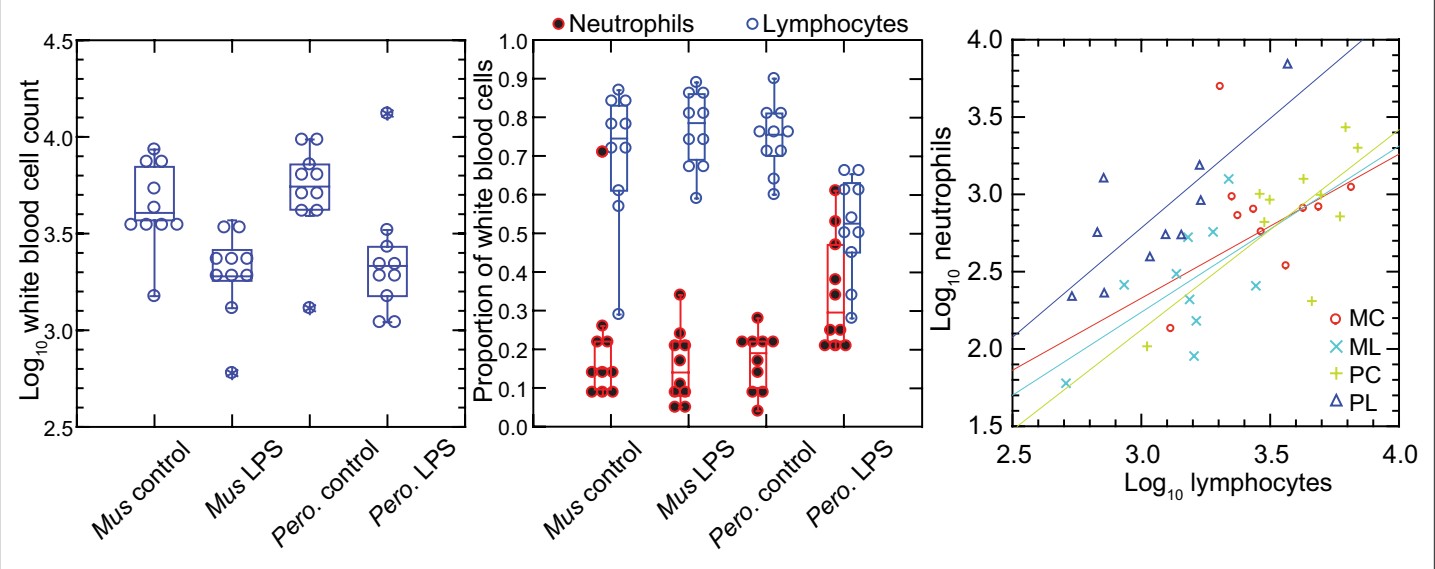

**Figure 1.** Total white blood cells, neutrophils, and lymphocytes of *Mus musculus* (M) and *Peromyscus leucopus* (P) with or without (control; C) treatment with 10 µg lipopolysaccharide (LPS; L) per g body mass 4 hr previous. The data are from *Table 1*. The box plots of left and center panels show values of individual animals and compiled median, quartiles, and extreme values. The linear regressions of the right panel are color-coded according to the species and treatment designations. The outlier value for a *M. musculus* control (MM17) was excluded from the linear regression for that group.

be higher in deermice at a mean 3.4 (2.1–4.7)% than mice at 1.2 (0.5–1.9)% under either condition (p=0.004).

In the *P. leucopus* experiment with a tenfold lower dose of LPS and a 12 hr duration, the mean (95% confidence interval) white blood cell count (x 10³) at termination 3.5 (2.5–4.5) in controls and 7.9 (6.0–9.7) in the LPS-treated (p=0.01). Even with the higher overall white blood cell count, the increase white cells was proportionately higher for neutrophils than for lymphocytes, as was seen in the deermice in the higher dose LPS experiment. The ratio of neutrophils-to-lymphocytes was 0.20 (0.07–0.32) in the controls and 0.38 (0.26–0.50) in the LPS-treated (p=0.10).

The higher neutrophil to lymphocyte ratio in the deermice exposed to LPS was consistent with the greater neutrophil activation noted by transcriptional analysis (*Balderrama-Gutierrez et al., 2021*). But many individual genes that constitute this and related gene ontology (GO) terms had transcription levels in the deermice that far exceeded a threefold difference in neutrophil counts. For some genes, the differences were a hundred or more fold, which suggested that the distinctive LPS transcriptional response profile between species was not attributable solely to neutrophil counts.

## Genome-wide expression in blood of deermice and mice

We used the respective transcript sets from the reference genomes for *P. leucopus* and *M. musculus* for deep coverage RNA-seq with paired-end ~150 nt reads (*Figure 2—source data 1* and *Figure 2—source data 2*). Principle component analyses (PCA) of the *P. leucopus* data and *M. musculus* data revealed that untreated controls had coherent profiles within each species (*Figure 2*). With the exception of one mouse, the LPS-treated *M. musculus* were also in a tight PCA cluster. In contrast, the LPS-treated deermice displayed a diversity of genome-wide transcription profiles and limited clustering.

For both species, the number of genes with higher expression with LPS exposure exceeded those with lower or unchanged expression. For *P. leucopus* and *M. musculus,* the mean fold-changes were 1.32 (1.29–1.35) and 1.30 (1.24–1.36), respectively (p=0.31). For GO term analysis, the absolute fold-change criterion was ≥2. Because of the ~3 fold greater number of transcripts for the *M. musculus* reference set than the *P. leucopus* reference set, application of the same false-discovery rate (FDR) threshold for both datasets would favor the labeling of transcripts as DEGs in *P. leucopus*. Accordingly, the FDR $p$ values were arbitrarily set at $<5 \times 10^{-5}$ for *P. leucopus* and $<3 \times 10^{-3}$ for *M. musculus* to provide approximately the same number of DEGs for *P. leucopus* (1154 DEGs) and *M. musculus* (1266 DEGs) for the GO term comparison.

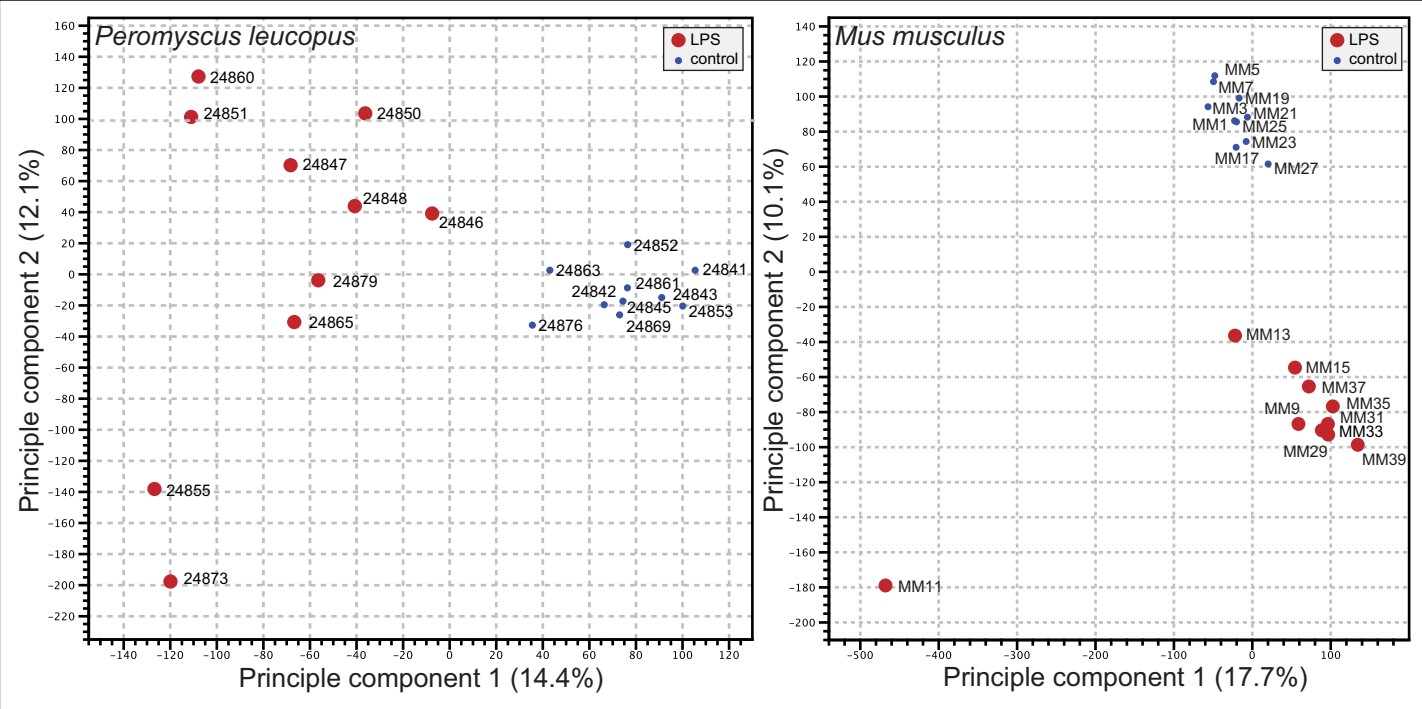

**Figure 2.** Principle component analysis of genome-wide RNA-seq data of *Peromyscus leucopus* or *Mus musculus* with or without (blue dot) treatment with LPS 4 hr previous (*Figure 2—source data 1* and *Figure 2—source data 2*). The individual animals listed in *Table 1* are indicated on the graphs. The insets indicate the size and color of the symbol for the experimental condition (LPS-treated or control).

The online version of this article includes the following source data for figure 2:

**Source data 1.** Genome-wide RNA-seq data as TPM values for *Peromyscus leucopus* treated with lipopolysaccharide or saline alone.

**Source data 2.** Genome-wide RNA-seq data as TPM values for *Mus musculus* treated with lipopolysaccharide or saline alone.

*Figure 3* shows the GO terms for the top 20 clusters by ascending p-value for up-regulated and down-regulated in *P. leucopus* and the corresponding categories for *M. musculus*. The up-regulated gene profile for *P. leucopus* featured terms associated with 'neutrophil degranulation', 'myeloid leukocyte activation', 'leukocyte migration', and 'response to molecule of bacterial origin'. Other sets of up-regulated genes for the deermice were 'negative regulation of cytokine production' and 'regulation of reactive oxygen species metabolic process'. None of these were among the top 20 up-regulated clusters for *M. musculus*. Indeed, 'leukocyte activation' and 'leukocyte migration' were GO terms for down-regulated DEGs in *M. musculus*. Distinctive GO terms for up-regulated genes distinguishing mice from deermice were 'response to virus', 'response to interferon-beta', 'response to interferon-gamma', 'response to protozoan', and 'type II interferon signaling'.

By arbitrary criterion of 100 for the top DEGs by ascending p value for each species, 24 genes were shared between species (*Figure 3—source data 3*). These included up-regulated *Bcl3, Ccl3, Cxcl1, Cxcl2, Cxcl3, Cxcl10, Il1rn*, and *Sod2*. Among the 100 mouse DEGs, 20 were constituents of GO terms 'response to virus' or 'response to interferon-beta' and only hree were members of GO term sets 'response to molecule of bacterial origin' or 'response to lipopolysaccharide'. In contrast, among the top 100 deermouse DEGs, there were only 2 associated with the virus or type 1 interferon GO terms, but 12 were associated with either or both of the bacterial molecule GO terms.

We confirmed the sex identification for each sample with sex-specific transcripts of *Xist* for females and *Ddx3y* for males (*Balderrama-Gutierrez et al., 2021*). For female and male *P. leucopus*, there were 5012 transcripts out of 54,466 in the reference set for which there were TPM values of ≥10 in at least one animal in each of the sexes under either condition . The comparison of females to males by fold-changes between LPS-treated and control animals revealed transcripts that were differentially expressed between sexes under LPS treatment (*Figure 4* and ). Some were down-regulated in one sex while unchanged in expression in the opposite sex. Of note in this category were different isoforms or variants of *Lilra6* (leukocyte immunoglobulin-like receptor, subfamily A, member 6), one of a family

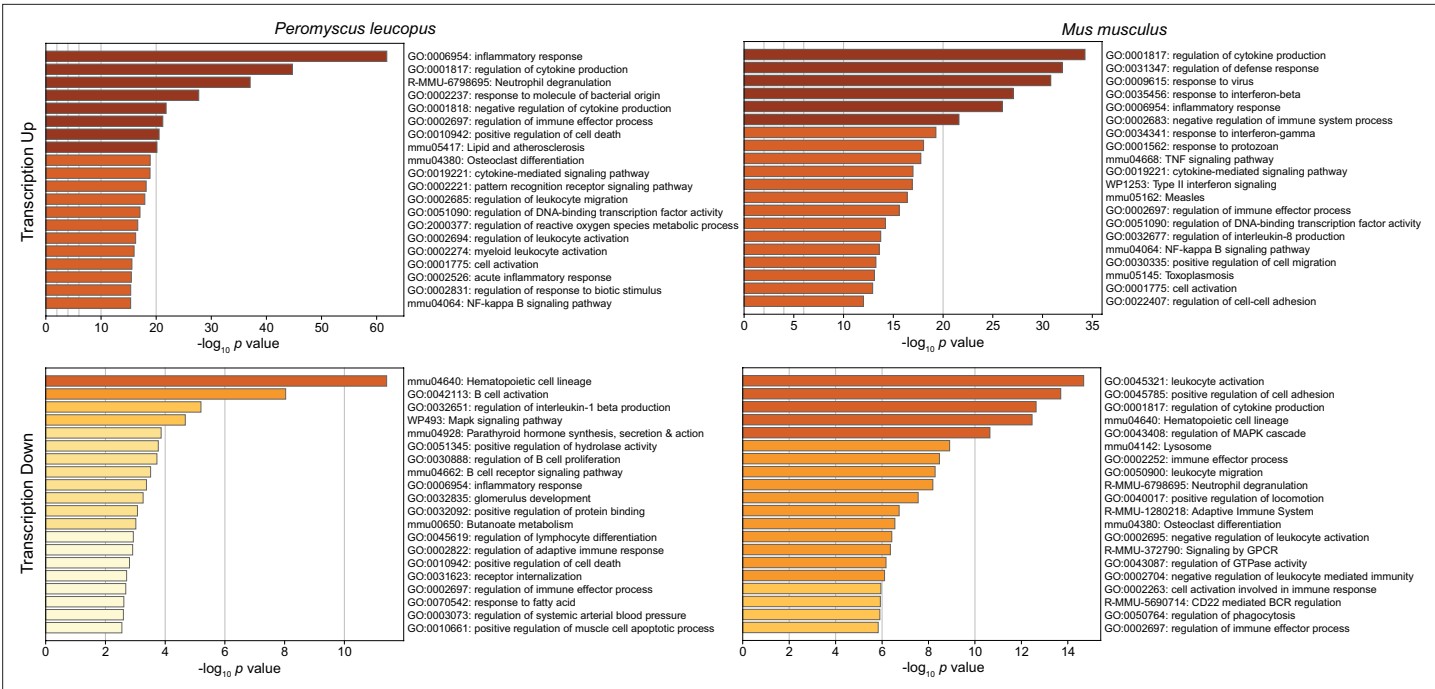

**Figure 3.** Gene Ontology (GO) term clusters associated with up-regulated genes (upper panels) and down-regulated genes (lower panels) of *Peromyscus leucopus* (left panels) and *Mus musculus* (right panels) treated with LPS in comparison with untreated controls of each species (*Figure 3—source data 1* and *Figure 3—source data 2*). The scale for the x-axes for the panels was determined by the highest -log₁₀ p values in each of the four sets. The horizontal bar color, which ranges from white to dark brown through shades of yellow through orange in between, is a schematic representation of the -log₁₀ p values.

The online version of this article includes the following source data for figure 3:

**Source data 1.** Differentially expressed gene analysis for *Peromyscus leucopus*.

**Source data 2.** Differentially expressed gene analysis for *Mus musculus*.

**Source data 3.** Comparison of differentially-expressed genes in genome-wide RNA-seq of blood of *Peromyscus leucopus* and *Mus musculus* with and without treatment with lipopolysaccharide.

of orphan receptors of myeloid cells (*Bashirova et al., 2014*). The opposite case was exemplified by the *Dnajc15* and *Hspa8* genes for two chaperones: DnaJ heat shock protein family (Hsp40) member C15 and heat shock protein 8, respectively. These were substantially lower in transcription in the LPS-treated females than in untreated animals, but little changed in LPS-treated males. Coordinates for some other genes, for example *Saa5* and *Cxcl2*, fell outside the prediction limits at the extreme end of up-regulation, but their vectors were within 20–25° of each other. While these and other sex-associated differences merit attention for future studies, overall they were not of sufficient number or magnitude in our view to warrant division by sex for the subsequent analyses, which had the aim of identifying differences applicable for both females and males.

## Targeted RNA seq analysis

The emerging picture was of *P. leucopus* generally responding to LPS exposure as if infected with an extracellular bacterial pathogen, including with activated neutrophils. While *M. musculus* animals of both sexes shared with *P. leucopus* some features of an antibacterial response, they also displayed type 1 and type 2 interferon type response profiles associated with infections with viruses and intra-cellular bacteria and parasites.

Going forward, the challenge for a cross-species RNA-seq was commensurability between anno-tated transcripts of reference sets. Orthologous genes can be identified, but mRNA isoforms and their 5′ and 3′ untranslated regions may not fully correspond. Accordingly, we limited targeted RNA-seq to protein coding sequences of mRNAs for the corresponding sets of *P. leucopus* and *M. musculus* sequences.

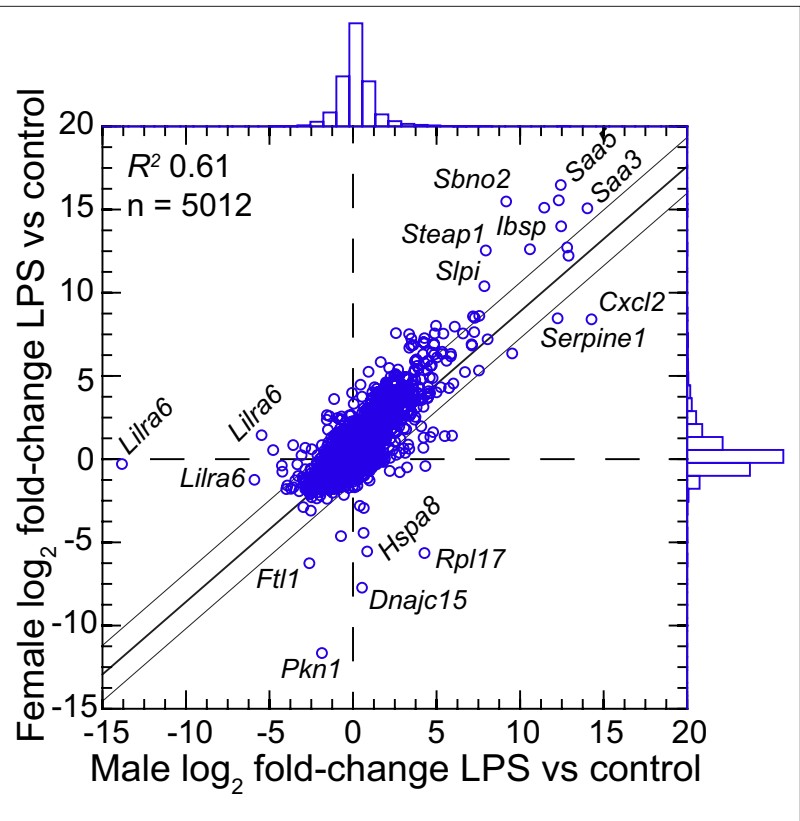

**Figure 4.** Scatter plot with linear regression of pairs of log$_2$-transformed mean fold-changes between LPS-treated and control *P.leucopus* by male and female sex (**Figure 4—source data 1** and **Figure 4—source data 2**). The coefficient of determination (R$^2$), the 95% upper and lower prediction limits for the regression line, and distributions of the values on the *x*- and *y*-axes are shown. Selected genes for which their *x-y* coordinates fall outside the limits of prediction are labeled. *Cxcl2, Ibsp, Saa3, Saa5, Sbno2, Serpine1, Slpi,* and *Steap1* were noted as up-regulated DEGs for the groups with both sexes (**Figure 3—source data 3**).

The online version of this article includes the following source data for figure 4:

**Source data 1.** Differentially expressed gene analysis by genome-wide RNA-seq of 20 female and 20 male *P. leucopus* treated with lipopolysaccharide or saline.

**Source data 2.** Comparison of female and male P. leucopus for mean fold-changes of LPS-treated to control animals for each of 5012 reference transcripts with a maximum TPM ≥10.

The 113 mRNA coding sequences, which are listed in Methods, were drawn from the identified DEGs for *P. leucopus* and *M. musculus* from the genome-wide RNA-seq. For cross-species normalization, we first evaluated three methods: (1) normalization using the ratio of mean total reads for all samples to total reads for a given sample, (2) the ratio of reads mapping to a given target transcript (i.e. the numerator) to the reads to transcripts of mitochondrial 12 S rDNA gene (i.e. the denominator), or (3) when the denominator instead was the myeloid cell marker CD45, or protein tyrosine phosphatase, receptor type C, encoded by *Ptprc*. In humans, mice, and hamsters, *Ptprc* is expressed by nucleated hematopoietic cells, and the protein CD45 is commonly used as a white cell marker for flow cytometry (*Schnizlein-Bick et al., 2002*). The coefficients of determination (R$^2$) between comparison pairs (e.g. normalization for total reads vs. normalization by *Ptprc* reads) within a species were ≥0.95 (*Figure 5*; *Figure 5—source data 1*). There was also little difference between the choice of *Ptprc* or 12 S rRNA transcripts as denominator with respect to cross-species comparisons of LPS-treated to control fold changes. The results indicated that the two normalization methods, one based on a mitochondrion gene and other on a chromosome gene, were commensurate. Given the widespread adoption of CD45 for flow cytometry, we chose *Ptprc* reads as denominator and as an adjustment for white cell numbers in the samples. Pearson correlation between log-transformed total white blood

| Genus Normalization | *Mus* reads $R^2$ | *Mus* 12S rDNA $R^2$ | *Mus* Ptprc $R^2$ | *Peromyscus* reads $R^2$ | *Peromyscus* 12S $R^2$ | *Peromyscus* Ptprc $R^2$ |
|---|---|---|---|---|---|---|
| *Mus* reads | 1.000 | | | | | |
| *Mus* 12S rDNA | 0.994 | 1.000 | | | | |
| *Mus* Ptprc | 0.969 | 0.945 | 1.000 | | | |
| *Peromyscus* reads | 0.394 | 0.371 | 0.426 | 1.000 | | |
| *Peromyscus* 12S | 0.416 | 0.396 | 0.446 | 0.986 | 1.000 | |
| *Peromyscus* Ptprc | 0.424 | 0.401 | 0.457 | 0.996 | 0.990 | 1.000 |

**Figure 5.** Comparison of three different methods for normalization for cross-species targeted RNA-seq. The normalization options were total reads for the same sample, unique reads for the mitochondrial 12 S rDNA, and unique reads for the *Ptprc* transcript encoding CD45. The 109 targets and all values for the analysis are in *Figure 5—source data 1*. For the upper panel, the coefficients of determination ($R^2$) were calculated for each of the pairs and for within each species and across species (columns C-G of *Source data 1*). The results of this analysis are in the matrix of the upper panel. The lower left panel compares in the same scatterplot the LPS to control fold-changes by either the 12 S or *Ptprc* normalization method, and the *Peromyscus leucopus* (P) result regressed on the *Mus musculus* (M) result for the same gene. The lower right panel of the figure is a scatterplot with linear regression and the $R^2$ value (columns I and J of *Source data 1*).

The online version of this article includes the following source data for figure 5:

**Source data 1.** Comparison of different methods for normalization (total reads, 12S rRNA or Ptprc) for cross-species targeted RNA-seq.

cell counts and normalized reads for Ptprc across 40 animals representing both species, sexes, and treatments was 0.40 (p=0.01).

*Figure 6* comprises plots of the log-transformed mean ratios for the 10 *P. leucopus* controls and 10 *M. musculus* controls and for the 10 *P. leucopus* and 10 *M. musculus* treated with LPS (*Figure 6—source data 1*). For untreated animals (left panel) there was high correlation and a regression coefficient of ~1 between the paired data for deermice and mice. *MT-Co1*, the gene for mitochondrial cytochrome oxidase 1 gene, and *S100a9*, which encodes a subunit of calprotectin, were comparably transcribed. But, there were other coding sequences that stood out for either their greater or lesser transcription in untreated deermice than mice. Two examples of greater expression were *Arg1* and *Mx2*, which encodes MX dynaminin-like GTPase 2, while two examples of lesser expression were *Mmp8*, the gene for matrix metalloprotease 8, and *Slpi*. There was low to undetectable transcription of *Nos2* and *Ifng*, the gene for interferon-gamma, in the blood of controls of both species.

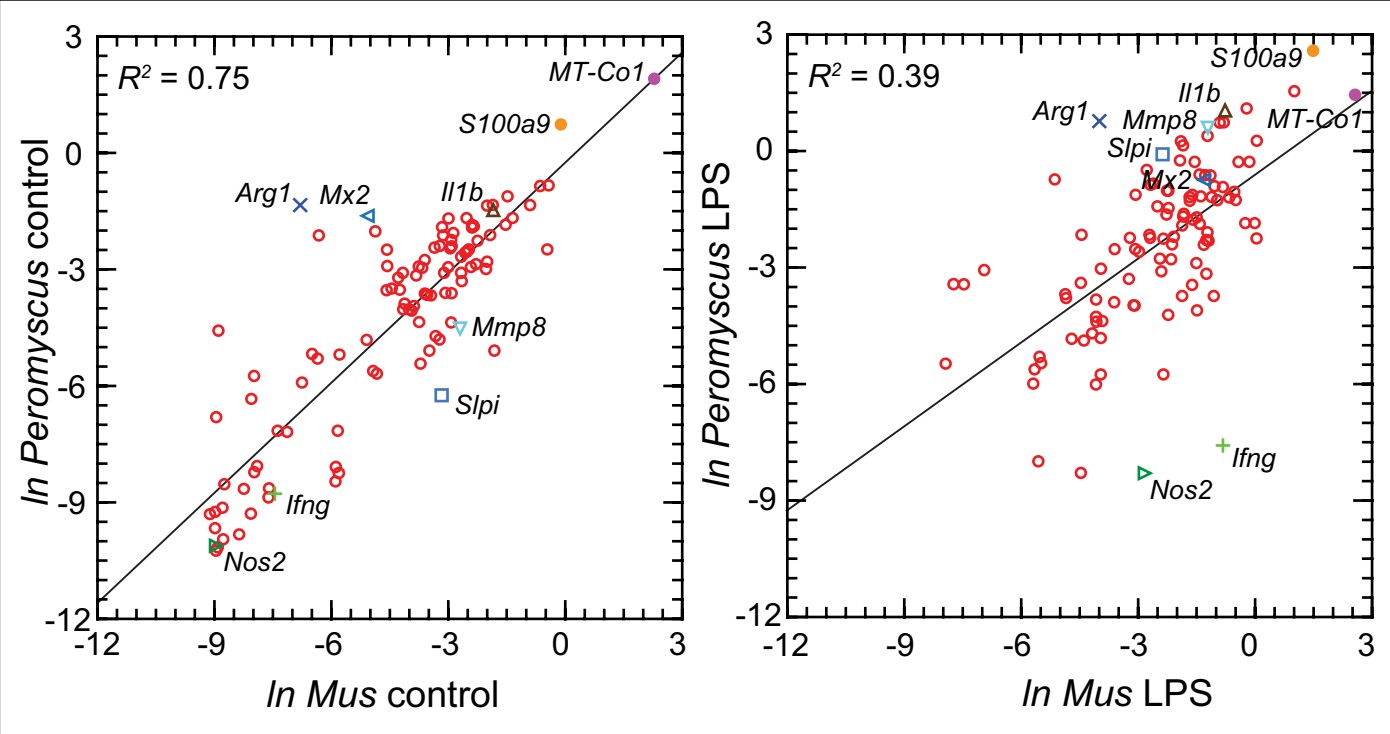

**Figure 6.** Scatter plots with linear regression of pairs of log-transformed (*ln*) normalized RNA-seq reads for selected coding sequences for control *Peromyscus leucopus* and *Mus musculus* (left panel) and LPS-treated *P. leucopus* and *M. musculus* (right panel) (*Figure 6—source data 1*). The $R^2$ values and selected genes (each with a different symbol) are indicated in each graph. Box plots for a selected 54 of these targets organized by functional characteristics are provided in *Figure 6—figure supplement 1*.

The online version of this article includes the following source data and figure supplement(s) for figure 6:

**Source data 1.** Natural logarithms of ratios of transcript reads of selected genes to Ptprc (Cd45) transcript reads in blood of *P. leucopus* or *M. musculus* with or without treatment with LPS by individual animal.

**Figure supplement 1.** Box plots of log-transformed normalized transcripts in whole blood for 54 genes of *Peromyscus leucopus* (P) or *Mus musculus* (M) that have been treated with LPS (L) or were saline-alone controls (C).

For the LPS-treated animals (right panel *Figure 6*) there was, as expected for this selected set, higher expression of the majority genes and greater heterogeneity among *P. leucopus* and *M. musculus* animals in their responses for represented genes. In contrast to the findings with controls, *Ifng* and *Nos2* had higher transcription in treated mice. In deermice the magnitude of difference in the transcription between controls and LPS-treated was less. A comparatively restrained transcriptional response in deermice was also noted for *Mx2*. On the other hand, there were greater fold-changes from baseline in *P. leucopus* than in *M. musculus* for *Mmp8, Slpi, S100a9*, and *Il1b*, the gene for interleukin-1 beta.

*Supplementary file 1* lists all the selected targets with the means and confidence intervals for the normalized values for controls and LPS-treated *M. musculus* and controls and LPS-treated *P. leucopus* (*Source data 1*). The fold-changes within each species and between treatments across species are given. The final column is the ratio of the fold-change between LPS to control in *P. leucopus* to the corresponding value for *M. musculus*. This along with the derived heat-map of these ratios, presented in the second column, indicates the genes for which there was little difference between species in their responses to LPS—either up or down—as well as those that were comparatively greater or lesser in one species or the other. Several of these genes are considered in other specific contexts below. Of note are the places of *Nos2* and *Ifng* at the bottom of the table, and *Il1b* near the top at position 20.

**Table 2.** RT-qPCR of blood of LPS-treated and control *Peromyscus leucopus* and *Mus musculus*.

| Blood mRNA source | Gene | Control mean copies (95% CI) | LPS mean copies (95% CI) | Fold difference (LPS/control) | t test p/ Mann-Whitney p |
|---|---|---|---|---|---|
| | *Gapdh* | 1.2 (0.40–3.7) x $10^5$ | 2.4 (1.1–5.2) x $10^5$ | 1.95 | 0.35/0.49 |
| | *Nos2* | 191 (141–260) | 138 (61–315) | 0.72 | 0.47/0.53 |
| *P. leucopus* | *Arg1* | 4.6 (3.1–6.9) x $10^3$ | 12.3 (3.2–47) x $10^3$ | 2.66 | 0.18/0.55 |
| | *Gapdh* | 6.1 (2.3–16.0) x $10^6$ | 1.8 (0.80–3.9) x $10^6$ | 0.29 | 0.06/0.02 |
| | *Nos2* | 101 (68–151) | 1891 (866–4130) | 18.6 | <0.00001/0.002 |
| *M. musculus* | *Arg1* | 27 (15–20) | 16 (8–34) | 0.59 | 0.29/0.45 |

The online version of this article includes the following source data for table 2:

**Source data 1.** RT-qPCR of Gapdh, Nos2, and Arg1 transcripts in blood of *P. leucopus* or *M. musculus* with or without treatment with LPS.

## 'Alternatively activated' macrophages and 'nonclassical' monocytes in *P. leucopus*

While we could not type single cells using protein markers, we could assess relative transcription of established indicators of different white cell subpopulations in whole blood. The present study, which incorporated outbred *M. musculus* instead of an inbred strain, confirmed the previous finding of differences in *Nos2*, the gene for inducible nitric oxide synthase, and *Arg1*, the gene for arginase 1, expression between *M. musculus* and *P. leucopus* (*Figure 5*; *Supplementary file 1*). Results similar to the RNA-seq findings were obtained with specific RT-qPCR assays for *Nos2* and *Arg1* transcripts for *P. musculus* and *M. musculus* (*Table 2*; *Table 2—source data 1*).

Low transcription of *Nos2* in both in controls and LPS-treated *P. leucopus* and an increase in Arg1 with LPS was also observed in another experiment for present study where the dose of LPS was 1 µg/g body mass instead of 10 µg/g and the interval between injection and assessment was 12 hr instead of 4 hr (*Table 3*; *Table 3—source data 1*).

In addition to the differences in *Nos2* and *Arg1* expression for typing macrophage and monocyte subpopulations, there are also the relative expressions of three other pairs of genes: (1) *Il12* and *Il10*, where a lower *Il12/Il10* transcription ratio is more characteristic of alternatively activated or M2 type *Murray, 2017*; *van Stijn et al., 2015*; (2) *Akt1* and *Akt2*, the genes for two proto-oncogene kinases, where the associations are *Akt1* with M2-type and *Akt2* with M1-type macrophages *Arranz et al., 2012*; *Vergadi et al., 2017*; and (3) CD14 and CD16, or low affinity immunoglobulin gamma Fc region receptor III, encoded by *Fcgr3*, where low expression of *Cd14* and high expression of *Fcgr3* is associated with 'non-classical' monocytes (*Narasimhan et al., 2019*). There is evidence that nonclassical monocytes can change to M2-type macrophages (*Italiani and Boraschi, 2014*).

These four relationships, which are presented as log-transformed transcription ratios for *Nos2/Arg1*, *Il12/Il10*, *Akt1/Akt2*, and *Cd14/Fcgr3*, are shown in *Figure 6*. We confirmed the difference between *P. leucopus* and *M. musculus* in the ratios of *Nos2/Arg1* and *Il12/Il10* [3] with outbred mice and normalization for white cells. In both species, the *Akt1/Akt2* ratio declined in LPS-treated animals, but for *P. leucopus* the ratio remained >1.0 even among LPS-treated animals, while in the blood of *M. musculus* the ratio was <1.0 at baseline and declined further in the LPS-treated animals.

An orthologous gene for Ly6C (*Bothwell et al., 1988*), a protein used for typing mouse monocytes and other white cells, has not been identified in *Peromyscus* or other Cricetidae family members. Therefore, expression of *Cd14* was compared with expression of the *Ly6c* alternative *Fcgr3*, which deermice and other cricetines do have. In mice, the *Cd14/Fcgr3* transcription ratio increased from baseline in the LPS group. In the deermice, the ratio in control animals was midway between the two groups of mice but there was a marked decrease in the LPS-treated deermice (*Figure 7*). This was not associated with a fall in the absolute numbers or percentages of monocytes in the blood of these animals (*Table 1*).

Taken together, the *Nos2/Arg1*, *Il12/Il10*, *Akt1/Akt2*, and *Cd14/Fcgr3* relationships document a disposition toward alternatively activated macrophages and nonclassical monocytes in *P. leucopus*

**Table 3.** Targeted RNA-seq of *Peromyscus leucopus* blood in 12 hr experiment with LPS dose of 1 µg/g.

| Gene (alternative name) | Control (n=3) mean (95% CI)* | LPS (n=3) mean (95% CI)* | Fold change | FDR p value[†] |
|---|---|---|---|---|
| *Akt1* | 220 (12–4202) | 514 (321–825) | 2.3 | 0.05 |
| *Akt2* | 145 (97–217) | 336 (236–478) | 2.3 | 0.04 |
| *Arg1* | 146 (58–367) | 2812 (273–28,925) | 19 | 0.018 |
| *Cd14* | 82 (12–569) | 914 (161–5197) | 11 | 0.05 |
| *Cd69* | 165 (87–310) | 68 (14–329) | 0.42 | 0.15 |
| ERV *env* | 25 (3–242) | 40 (7–224) | 1.6 | 0.86 |
| ERV *gag-pol* | 3085 (132–14,695) | 2768 (533–1278) | 0.9 | 0.66 |
| *Fcgr3* | 40 (25–66) | 841 (533–1278) | 21 | 0.008 |
| *Gapdh* | 7176 (3504–15,142) | 23811 (5827–97,306) | 3.3 | 0.07 |
| *Gbp4* | 97 (6–1551) | 439 (33–5819) | 4.5 | 0.05 |
| *Ifit1* | 367 (51–2663) | 1373 (374–5047) | 3.7 | 0.05 |
| *Ifng* | 0 (0–0) | 0 (0–0) | . | . |
| *Il1b* | 258 (18–3652) | 1432 (183–11,220) | 5.6 | 0.1 |
| *Irf7* | 121 (93–157) | 11405 (530–245,616) | 94 | 0.003 |
| *Isg15* | 429 (184–1001) | 19505 (11140–34,152) | 45 | 0.005 |
| *Mx2* | 157 (73–341) | 1310 (323–5315) | 8.3 | 0.04 |
| *Nos2* | 0 (0–0) | 0 (0–0) | . | . |
| *Oas1* | 65 (31–138) | 1458 (367–5795) | 22 | 0.03 |
| *Rigi* (*Ddx58*) | 38 (3–504) | 173 (35–845) | 4.5 | 0.07 |
| *Saa3* | 2 (0–250) | 6683 (1494–29,896) | 3372 | 0.03 |
| *Slpi* | 6 (1-46) | 779 (326–1864) | 123 | 0.03 |
| *Sod2* | 180 (54–607) | 3406 (698–16,633) | 19 | 0.03 |

*Mean unique reads for given gene normalized for reads for Ptprc (Cd45) gene for a sample. The 95% confidence intervals (CI) are asymmetric. Actual [gene]/Ptprc ratios are x $10^{-3}$ (**Source data 1**).
[†]FDR, false discovery rate p value.

The online version of this article includes the following source data for table 3:

**Source data 1.** Targeted RNA-seq of blood of *P. leucopus* 12 h after treatment with LPS (1 µg/g) or saline.

both before and after exposure to LPS. This contrasts with profiles consistent with a predominance of classically activated macrophages and classical monocytes in mice.

## Interferon-gamma and interleukin-1 beta dichotomy between deermice and murids

For mice the *Ifng* transcript was one of the top ranked DEGs by both fold-change and adjusted p value by genome-wide RNA-seq (**Supplementary file 1**). In contrast, for *P. leucopus Ifng* was far down the list, and the comparably ranked DEG instead was *Il1b*. This inversion of relationships between two pro-inflammatory cytokines was confirmed by analysis of the individual animals of both species (**Figure 8**). There was little or no detectable transcription of *Ifng* in the blood of deermice in which *Il1b* expression was high. There was also scant to no transcription of *Ifng* in the blood of *P. leucopus* 12 hr after injection of LPS (**Table 3**).

The up-regulation of the interferon-gamma gene within 4 hr of exposure to LPS was not limited to the species *M. musculus*. In an experiment with the rat *R. norvegicus*, we used two different LPS doses (5 µg/g and 20 µg/g), but the same 4 hr endpoint and whole blood as the sample. Both groups

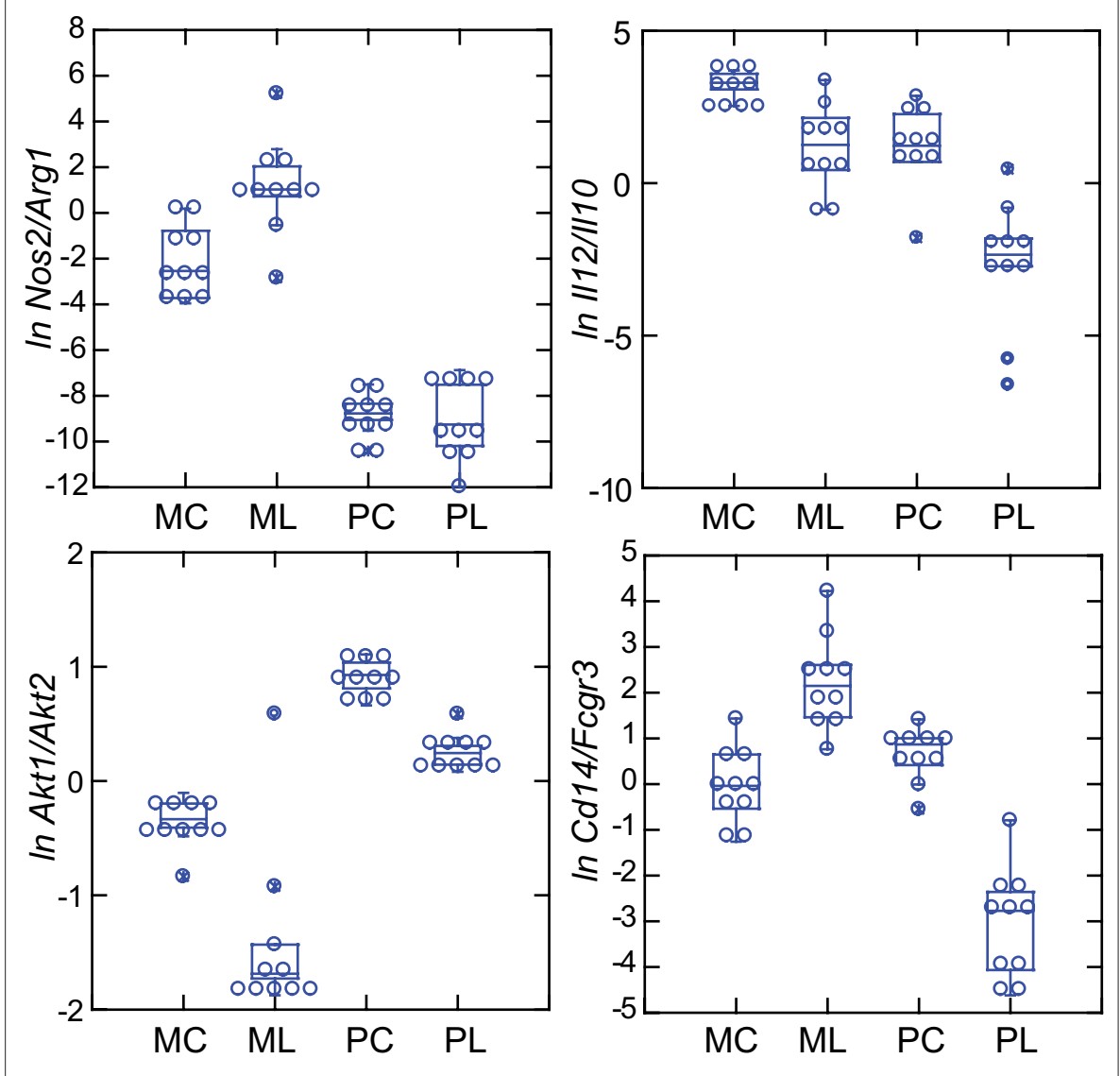

**Figure 7.** Box plots of natural log (*ln*)-transformed ratios of four pairs of gene transcripts from targeted RNA-seq analysis of blood of *Peromyscus leucopus* (P) or *Mus musculus* (M) with (L) or without (C) treatment with LPS. The values are from **Source data 1**. Upper left, *Nos2/Arg1*; upper right, *Il12/Il10*; lower left, *Akt1/Akt2*; lower right, *Cd14/Fcgr3*.

of LPS-treated rats had lowered total white blood cells and, like the mice, lower neutrophil-to-lymphocyte ratios compared to controls (**Table 4**; **Table 4—source data 1**). There were also elevations of interferon-gamma, interleukin-6, and interleukin-10 proteins from undetectable levels in the blood of the treated rats. The values for rats receiving 5 μg/g or 20 μg/g doses were similar, so these groups were combined. By targeted RNA-seq, there were 24 x fold-changes between the LPS-treated rats and control rats for *Ifng* and *Nos2* but only ~3 x fold-change for *Il1b* (**Table 4**).

Given these findings, we asked why the interferon-gamma response observed in CD-1 mice and rats here was not as pronounced in BALB/c mice (**Balderrama-Gutierrez et al., 2021**). Accordingly, we used the RNA-seq reads obtained from the prior study in combination with the reads of the present study and carried out targeted RNA-seq (**Figure 9**). The BALB/c inbred mice had, like the CD-1 mice, modest elevations of *Il1b* transcription. *Ifng* expression was also elevated in the BALB/c animals but not to the degree noted in CD-1 mice or rats. One explanation is an inherent difference of BALB/c mice from other strains in their lower interferon-gamma response to LPS (**Kuroda et al., 2002**; **Soudi et al., 2013**).

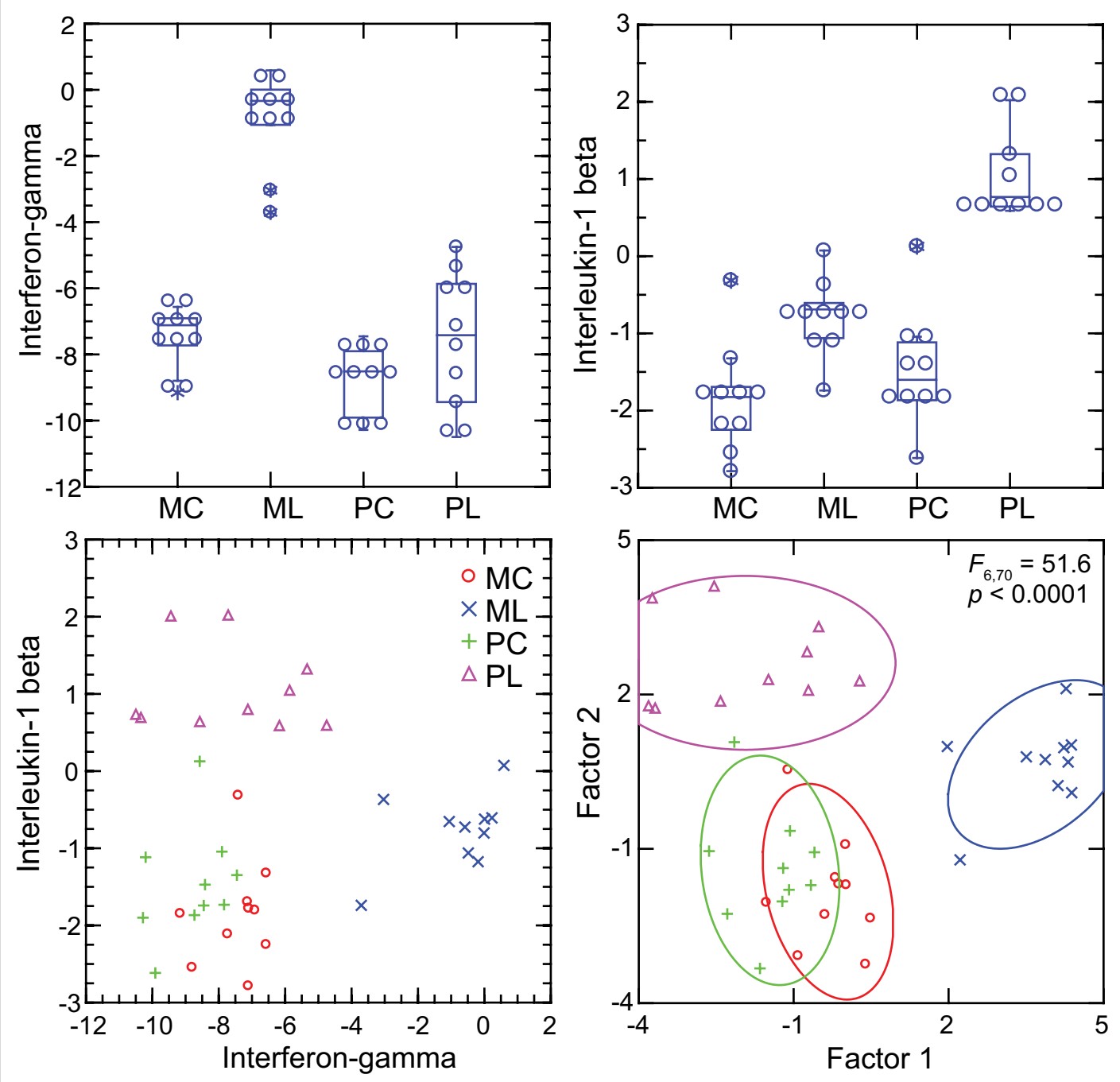

**Figure 8.** Transcripts of genes for interferon-gamma and interleukin-1 beta by targeted RNA-seq of the blood of *Peromyscus leucopus* (P) or *Mus musculus* (M) with (L) or without (C) treatment with LPS. The top panels are box plots of the individual values. The lower left panel is a scatter plot of *Il1b* on *Ifng* transcription values. The lower right panel is a Discriminant Analysis of these pairs of values where Factor 1 corresponds to *Ifng*, and Factor 2 corresponds to *Il1b*. Values for analysis are from **Source data 1**.

## Interferon-gamma and inducible nitric oxide synthase

Interferon-gamma is a determinant of *Nos2* expression (**Lowenstein et al., 1993**; **Salkowski et al., 1997**). So, the scant transcription of *Ifng* in *P. leucopus* conceivably accounted for the low expression of *Nos2* in that species. The analysis shown in upper left panel of **Figure 10** shows a tight correlation between the levels of transcription of *Ifng* and *Nos2* for both species and both experimental conditions (**Figure 10—figure supplement 1**). A significant correlation was also observed for the combined

**Table 4.** Hematology, cytokines, and targeted RNA-seq of LPS-treated and control *Rattus norvegicus*.

| Variable | Control (n=5) mean (95% CI)* | LPS (n=11) mean (95% CI)*,† | Fold change | FDR p value |
|---|---|---|---|---|
| Hematology | | | | |
| Hematocrit (%) | 48 (46–50) | 48 (46–49) | 1.0 | 1E+00 |
| White blood cells | 7660 (7200–8310) | 4980 (2020–7940) | 0.75 | 2E-01 |
| Neutrophils | 3680 (3260–4090) | 1410 (520–2290) | 0.38 | 4E-03 |
| Lymphocytes | 3170 (3010–3330) | 2830 (1230–4430) | 0.89 | 8E-01 |
| Neutrophil/lymphocyte | 1.17 (1.01–1.33) | 0.49 (0.44–0.55) | 0.42 | 7E-09 |
| Blood cytokines (pg/ml) | | | | |
| Interleukin-6 | 0 (0–0) | 36933 (21676–52190) | . | 5E-15 |
| Interleukin-10 | 9 (1-17) | 640 (477–802) | 71 | 2E-09 |
| Interferon-gamma | 0 (0–0) | 9091 (7126–11056) | . | 9E-20 |
| Targeted RNA-seq | | | | |
| *Akt1* | 35.8 (29.8–42.9) | 24.3 (21.5–27.5) | 0.68 | 4E-03 |
| *Akt2* | 50.8 (44.9–57.4) | 109 (95.2–125) | 2.2 | 9E-06 |
| *Arg1* | 0.04 (0.02–0.08) | 0.21 (0.10–0.44) | 4.8 | 2E-02 |
| *Cd14* | 7.7 (5.3–11.2) | 43.4 (29.6–63.6) | 5.6 | 9E-05 |
| *Cd177* | 0.89 (0.43–1.8) | 190 (143–251) | 213 | 2E-10 |
| *Cd3* | 32.3 (27.7–37.6) | 21.9 (18.5–25.8) | 0.68 | 1E-02 |
| *Cd69* | 17.9 (16.7–19.2) | 58.9 (49.1–70.7) | 3.3 | 9E-07 |
| *Cgas* | 1.5 (1.3–1.6) | 12.8 (10.8–15.1) | 8.7 | 3E-10 |
| *Cxcl10* | 0.43 (0.34–0.56) | 130 (93.7–181) | 302 | 1E-11 |
| *Dhx58* | 7.1 (6.7–7.6) | 113 (97.3–130) | 15.8 | 3E-12 |
| ERV *env* | 8.4 (7.5–9.3) | 713 (624–815) | 85.3 | 1E-14 |
| ERV *gag-pol* | 6.0 (5.4–6.7) | 506 (449–570) | 84.3 | 7E-15 |
| *Fcgr2a* | 114 (87.0–150) | 764 (631–925) | 6.7 | 4E-08 |
| *Fcgr2b* | 32.5 (24.6–43.1) | 161 (127–204) | 4.9 | 2E-06 |
| *Fcgr3* | 15.2 (13.6–17.0) | 13.9 (11.8–16.3) | 0.91 | 5E-01 |
| *Gapdh* | 327 (237–451) | 1643 (1385–1949) | 5.0 | 2E-07 |
| *Gbp4* | 35.7 (33.6–38.0) | 269 (237–306) | 7.5 | 3E-11 |
| *Ifih1* | 18.7 (17.1–20.4) | 165 (149–184) | 8.8 | 2E-12 |
| *Ifit1* | 102 (70.0–147) | 756 (677–844) | 7.4 | 3E-09 |
| *Ifng* | 0.47 (0.32–0.67) | 10.3 (6.4–16.5) | 22.1 | 1E-06 |
| *Il10* | 0.12 (0.07–0.21) | 4.5 (3.4–5.8) | 38.2 | 5E-09 |
| *Il12* | 0.07 (0.04–0.11) | 3.2 (2.0–4.9) | 45.9 | 7E-08 |
| *Il1b* | 58.6 (39.7–86.4) | 618 (503–760) | 10.6 | 2E-08 |
| *Il6* | 0.06 (0.05–0.08) | 4.4 (2.9–6.6) | 70.9 | 6E-09 |
| *Irf7* | 44.8 (36.9–54.3) | 443 (372–528) | 9.9 | 6E-10 |
| *Isg15* | 15.6 (13.1–18.7) | 624 (534–729) | 39.9 | 6E-13 |

*Table 4 continued on next page*

*Table 4 continued*

| Variable | Control (n=5) mean (95% CI)* | LPS (n=11) mean (95% CI)*,† | Fold change | FDR p value |
|---|---|---|---|---|
| *Itgam* | 66.3 (52.5–83.7) | 208 (161–269) | 3.1 | 9E-05 |
| *Mmp8* | 75.2 (50.4–112) | 519 (438–615) | 6.9 | 7E-08 |
| *Mx2* | 40.7 (35.9–46.2) | 900 (780–1039) | 22.1 | 1E-12 |
| *Nos2* | 32.4 (17–60.6) | 2990 (2491–3589) | 92.4 | 1E-10 |
| *Oas1* | 23.1 (18.8–28.4) | 151 (140–164) | 6.6 | 3E-11 |
| *Rigi (Ddx58)* | 8.6 (7.8–9.4) | 151 (135–168) | 17.6 | 1E-13 |
| *S100a9* | 298 (190–466) | 2884 (2269–3666) | 9.7 | 2E-07 |
| *Saa1* | 0.60 (0.49–0.73) | 699 (552–884) | 1167 | 4E-14 |
| *Slpi* | 20.2 (13.1–31.3) | 262 (197–347) | 12.9 | 1E-07 |
| *Sod2* | 63.8 (51.1–79.6) | 901 (759–1070) | 14.1 | 2E-10 |
| *Tlr4* | 5.3 (4.6–6.0) | 20.1 (17.2–23.6) | 3.8 | 8E-08 |
| *Tnf* | 1.1 (0.63–1.9) | 78.8 (62.6–99.1) | 72.8 | 2E-10 |
| Ratios | | | | |
| *Akt1/Akt2* | 0.70 (0.65–0.76) | 0.22 (0.19–0.25) | 0.31 | 2E-08 |
| *Cd14/Fcgr3* | 0.51 (0.34–0.76) | 3.14 (2.26–4.36) | 6.16 | 1E-05 |
| *IL12/IL10* | 0.54 (0.22–1.37) | 0.70 (0.51–0.95) | 1.30 | 5E-01 |
| *Nos2/Arg1* | 741 (403–1362) | 14244 (7615–26646) | 19.2 | 5E-05 |

*For targeted RNA-seq it is mean unique reads for given gene normalized for reads for Ptprc (Cd45) gene for a sample. The 95% confidence intervals (CI) are asymmetric. Actual [gene]/Ptprc ratios are x 10–3.
†The results for rats receiving 5 µg/g (n=6) and 20 µg/g (n=5) were combined.

The online version of this article includes the following source data for table 4:

**Source data 1.** Targeted RNA-seq of blood of Rattus norvegicus with or without treatment with LPS and with normalization by Ptprc transcripts.

**Source data 2.** Differentially expressed genes of genome-wide RNA-seq of blood of *R. norvegicus* with and without treatment with LPS.

set of animals between the ratios of *Nos2* to *Arg1* on *Ifng* to *Il1b* (upper right panel), an indication of co-variation between *Ifng* expression and macrophage polarization.

The plausible sources of interferon-gamma mRNA in whole blood are T-cells, Natural Killer cells, and Type 1 Innate Lymphoid Cells **Quatrini et al., 2017**. A DEG for *M. musculus* by both genome-wide and targeted RNA-seq (**Supplementary file 1**) was *Cd69*, which encodes a C-type lectin protein and an early activation antigen for these cells **Heinzelmann et al., 2000**. In *P. leucopus*, transcription of *Cd69* occurred in the blood of control *P. leucopus*, but it was the same or only marginally different for the LPS-treated animals (lower left panel). In contrast, in *M. musculus* the baseline transcription of *Cd69* was below that of *P. leucopus*, while in the LPS-treated mice it was many fold higher. In mice, transcripts for *Cd69* correlated tightly with *Ifng* transcription, but in the deermice there was little correlation between *Cd69* and *Ifng* expression at those low levels (lower right panel).

The findings are consistent with CD69-positive cells being a source of Ifng in mice. *Cd69* transcription was comparatively higher in control deermice than in control mice, so we presume that deermice have CD69-positive cells at baseline. One explanation then for the comparatively few *Ifng* transcripts in the deermice after LPS is a diminished responsiveness of these cells. *Tlr4* expression increased ~threefold more in *P. leucopus* than in *M. musculus* after LPS (**Supplementary file 1**), but the magnitude of the decline in expression of *Cd14* in deermice than mice was even greater. CD14 is required for LPS-stimulated signaling through surface TLR4 **Mazgaeen and Gurung, 2020**, and, as

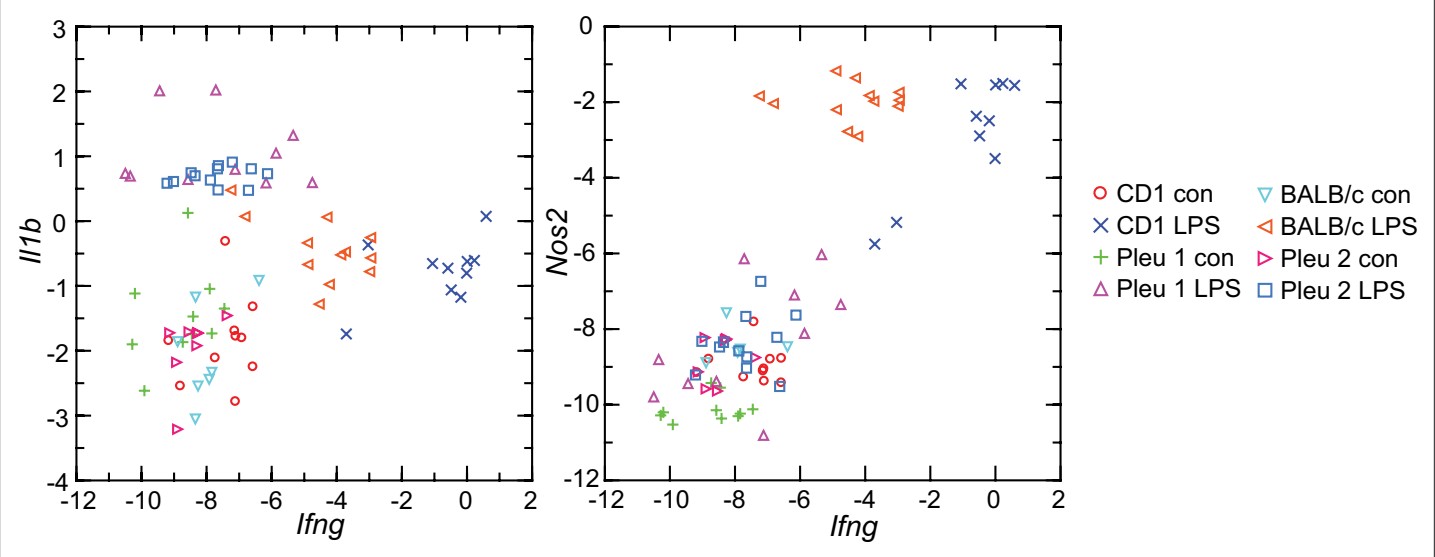

**Figure 9.** Scatter plots of log-transformed normalized transcripts of genes for interleukin-1 beta (*Il1b*; left panel) or nitric oxide synthase 2 (*Nos2*; right panel) on interferon-gamma (*Ifng*) of blood of *Peromyscus leucopus* (Pleu) or *Mus musculus* (outbred CD-1 and inbred BALB/c) with (LPS) or without (con) treatment with lipopolysaccharide 4 hr previously. The data are from the present study (Pleu 2 and CD-1) (***Source data 1***) and from the study of *Balderrama-Gutierrez et al., 2021* (Pleu 1 and BALB/c) (***Figure 9—source data 1***).

The online version of this article includes the following source data for figure 9:

**Source data 1.** Targeted RNA-seq of blood with normalization by Ptprc of LPS-treated and control *P. leucopus* and BALB/c *M. musculus* reported in Balderrama-Gutierrez et al.

such, its decreased availability for this signaling pathway is a possible explanation for the moderated response to LPS in *P. leucopus*.

## Interferon-stimulated genes and RIG-I-like receptors

As noted, GO terms differentiating mice from deermice included 'response to interferon-beta' and 'response to virus' (***Figure 3***). There was also the example of *Mx2*'s product, an ISG with antiviral activity on its own, that showed a greater fold-change from baseline in mice than in deermice (***Figure 5***). Five other ISGs (and encoding genes)—guanylate binding protein 4 (*Gbp4*), interferon-induced protein with tetratricopeptide repeat (*Ifit1*), interferon regulatory factor 7 (*Irf7*), ubiquitin-type modifier ISG15 (*Isg15*), and 2'–5' oligoadenylate synthase 1 A (*Oas1a*)—had higher transcription in all the LPS-treated animals. But the magnitude of fold change was less in the deermice, ranging from 6–25% of what it was in the LPS group of mice (***Supplementary file 1***).

The up-regulation of these ISGs was evidence of an interferon effect, but transcripts for interferon-1 beta (*Ifnb*) or -alpha (*Ifna*) themselves were scarcely detectable in deermice or mice in the blood under either condition. We then considered pattern recognition receptors (PRR) that might be part of a signaling pathway leading to ISG expression. Among the DEGs from the genome-wide analyses were genes for four cytoplasmic PRRs: (1) *Rigi* (formerly called *Ddx58*), which encodes the RNA helicase retinoic acid-inducible I (RIG-I); (2) *Ifih1*, which encodes interferon induced with helicase C domain 1, also known as MDA5 and a RIG-I-like receptor; (3) *Dhx58*, which encodes LGP2, another RIG-I-like receptor; and (4) *Cgas*, which encodes cyclic GMP-AMP synthase, part of the cGAS-STING sensing pathway.

All four cytoplasmic PRRs were upregulated in the blood of LPS-treated mice and deermice (***Supplementary file 1***). But, again, for each of them the magnitude of fold change was less by 50–90% in treated *P. leucopus* than in *M. musculus*. The coefficients of determination for the six ISGs and the four PRRs are provided in ***Figure 11***. For most of the pairs there was evidence of covariation across all 40 animals. When the correlation was low across all the data, for example between the ISG gene *Mx2* and the PRR gene *Rigi* or ISG genes *Mx2* and *Gbp4*, it was high within a species.

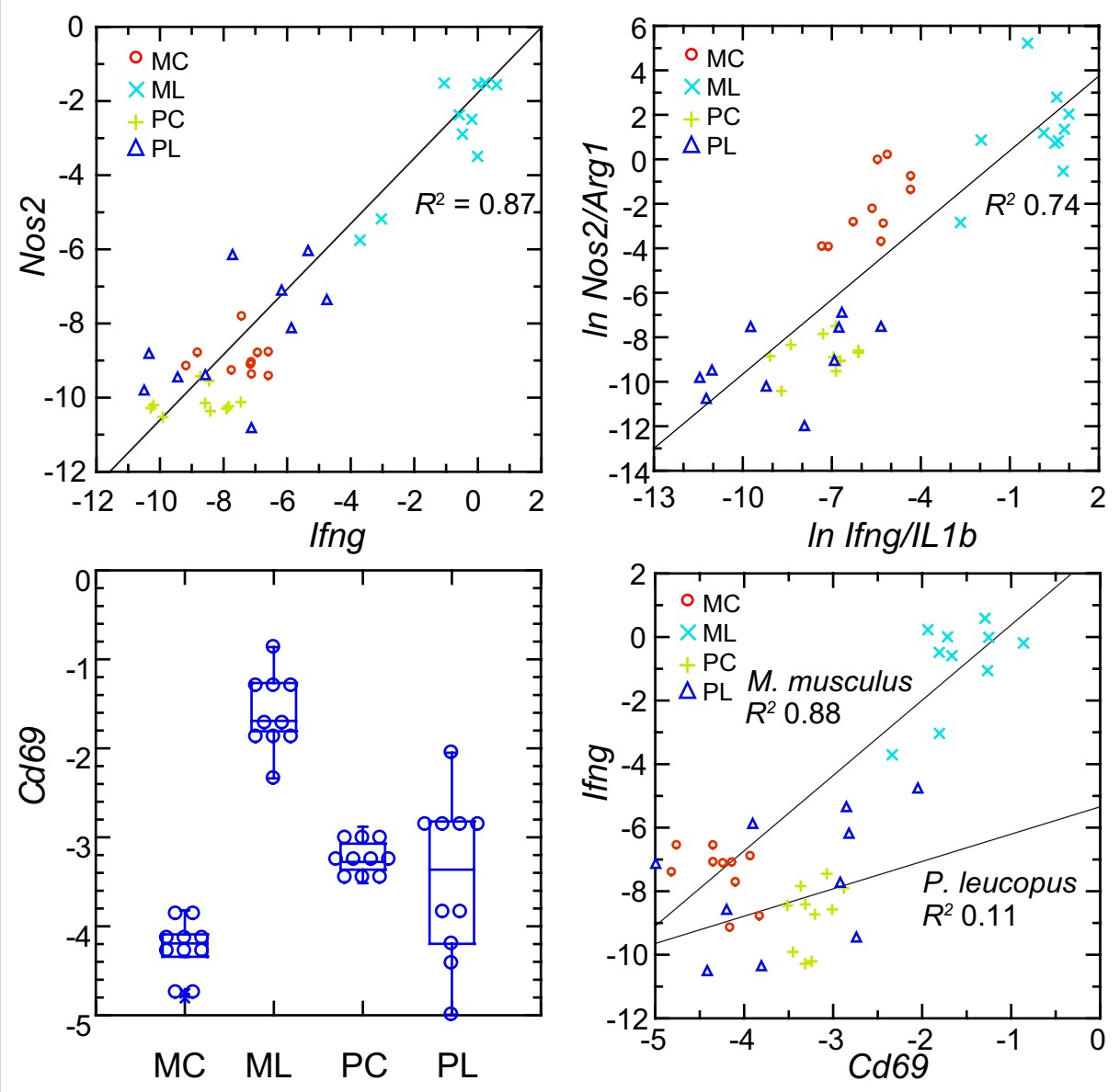

**Figure 10.** Normalized transcripts of *Nos2*, *Ifng*, and *Cd69* in targeted RNA-seq analysis of blood of *Peromyscus leucopus* (P) or *Mus musculus* (M) with (L) or without (C) treatment with LPS. Upper left: scatter plot of individual values for *Nos2* on *Ifng* with linear regression curve and coefficient of determination ($R^2$). Upper right: linear regression with $R^2$ of natural logarithms (*ln*) of *Nos2/Arg1* on *Ifng/Il1b*. Lower left: Box plots of individual values of normalized transcripts of *Cd69*. Lower right: Scatter plot of *Ifng* on *Cd69* transcription with separate regression curves and $R^2$ values for *M. musculus* and *P. leucopus*. Values for analysis are in **Source data 1**. Box plots for *Nos2* and *Arg1* are provided in **Figure 10—figure supplement 1**, and box plots for *Ifng* and *Il1b* are provided in **Figure 8**.

The online version of this article includes the following figure supplement(s) for figure 10:

**Figure supplement 1.** Box plots of log-transformed normalized transcripts of blood for nitric oxide synthase 2 (*Nos2*) and arginase 1 (*Arg1*) genes of *Peromyscus leucopus* (P) or *Mus musculus* (M) that have been treated with LPS (L) or were saline-alone controls (C).

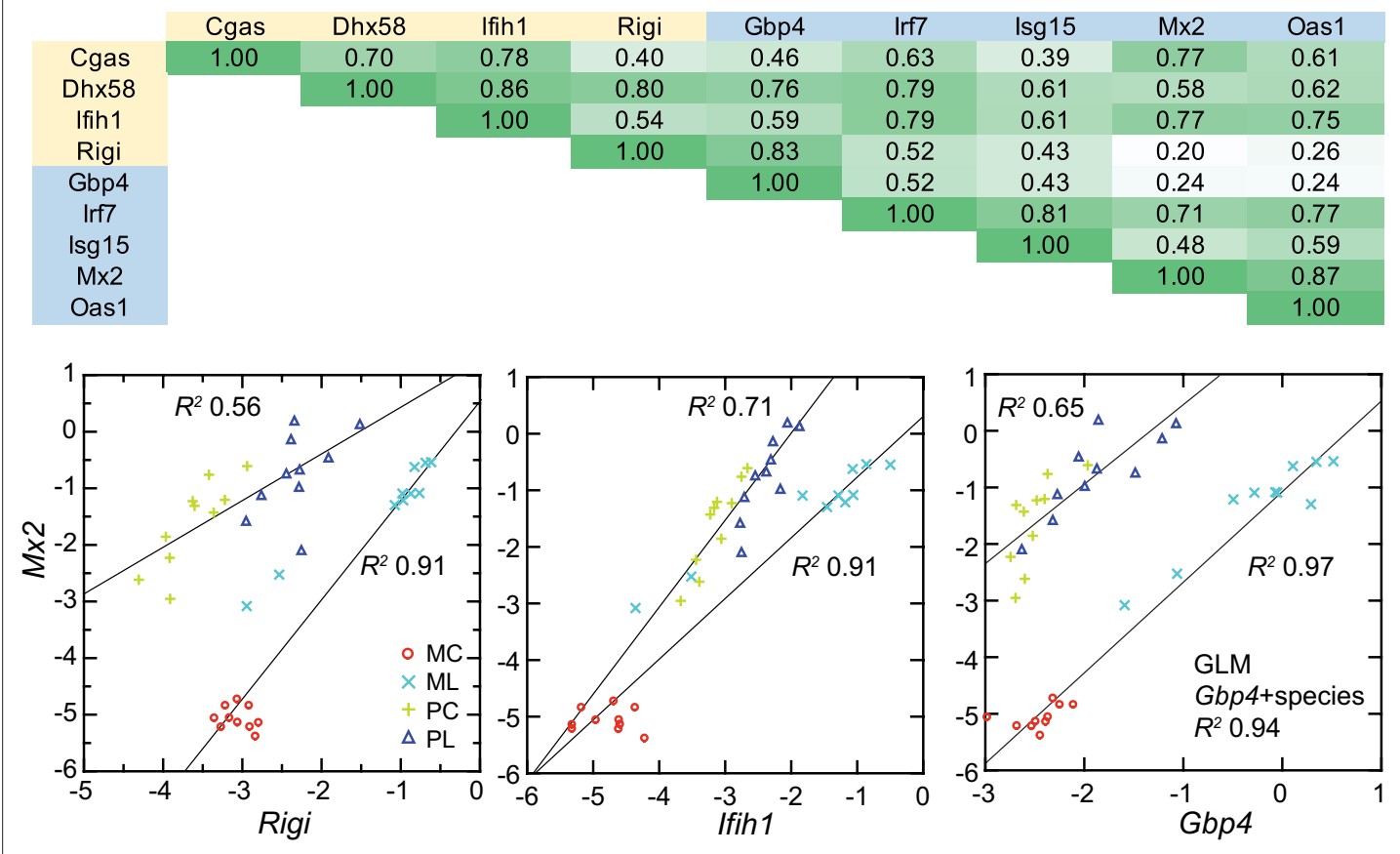

**Figure 11.** Co-variation between transcripts for selected PRRs and ISGs in the blood of *Peromyscus leucopus* (P) or *Mus musculus* (M) with (L) or without (C) LPS treatment. Top panel: matrix of coefficients of determination (R²) for combined *P. leucopus* and *M. musculus* data. PRRs are indicated by yellow fill and ISGs by blue fill on horizontal and vertical axes. Shades of green of the matrix cells correspond to R² values, where cells with values less than 0.30 have white fill and those of 0.90–1.00 have deepest green fill. Bottom panels: scatter plots of log-transformed normalized *Mx2* transcripts on *Rigi* (left), *Ifih1* (center), and *Gbp4* (right). The linear regression curves are for each species. For the right-lower graph the result from the General Linear Model (GLM) estimate is also given. Values for analysis are in ***Source data 1***; box plots for *Gbp4*, *Irf7*, *Isg15*, *Mx2*, and *Oas1* are provided in ***Figure 11—figure supplement 1***.

The online version of this article includes the following figure supplement(s) for figure 11:

**Figure supplement 1.** Box plots of log-transformed normalized transcripts for six interferon-stimulated genes of *Peromyscus leucopus* (P) or *Mus musculus* (M) that have been treated with LPS (L) or were saline-alone controls (C).

These findings were evidence that pathways in *P. leucopus* for PRR signaling and ISG expression functioned similarly to those in *M. musculus* but differed under these experimental conditions in magnitude of the changes, being more moderate in the deermice.

## Endogenous retroviruses in deermice, mice, and rats after LPS exposure

The six ISGs are nonexclusive consequences of activity of type 1 interferons. What we could document was the association of transcription of the genes for the cytoplasmic PPRs, including RIG-I, and the ISGs in both species, as well as the distinction between deermice in the magnitude of the responses of both PRRs and ISGs. These findings led us to ask could be a pathogen-associated molecular pattern (PAMP) for signaling pathways leading to expression of type 1 interferons.

One of these is endogenous retroviruses (ERV). The activity of these diverse, abundant, and pervasive elements have been recognized as one of the drivers of innate immune responses to a microbe (***Hurst and Magiorkinis, 2015***; ***Lima-Junior et al., 2021***; ***Rangel et al., 2022***). Our attention was drawn to ERVs by finding in the genome-wide RNA-seq of LPS-treated and control rats. Two of the three highest scoring DEGs by FDR *p* value and fold-change criteria were a *gag-pol* transcript

(GenBank accession XM_039101019.1) for an ERV polyprotein and an *env* transcript (XM_039113367) for an envelope (Env) protein that is similar to that of murine leukemia viruses (MLV) (*Table 4*; *Table 4—source data 2*).

We returned to the mouse and deermouse data. There were four MLV-type or other ERV *env* transcripts among the 1266 genome-wide RNA-seq DEGs for *M. musculus*. But, there was no transcripts for an ERV Env protein annotated as such among the 1154 DEGs identified for *P. leucopus* (*Figure 3—source data 3*). One possible explanation for the difference was an incomplete annotation of the *P. leucopus* genome. We took three approaches to rectify this. The first was to examine the DEGs for *P. leucopus* that encoded a polypeptide ≥200 amino acids and was annotated for the genome as 'uncharacterized'. A search of both the virus and rodent proteins databases with these candidates identified two that were homologous with *gag* and *pol* genes of mammalian ERVs.

For a second approach, we carried out a de novo transcript assembly of mRNA reads from blood of LPS-treated and control *P. leucopus* and used the resultant contigs as the reference set for RNA-seq analysis. This identified two contigs that were measurably transcribed in the blood, differentially expressed between conditions, and homologous to ERV sequences. One would encode an Env protein that was identical to a *P. leucopus* coding sequence (XM_037209467) for a protein annotated as 'MLV-related proviral Env protein'. The second was a *gag-pol* sequence that was near-identical to a *gag-pol* transcript (XM_037208848) identified by the first approach.

The third approach was to scan the *P. leucopus* genome for nonredundant sequences, defined as <95% identity, that were homologous with ERV *gag-pol* sequences, which are not typically annotated because of masking for repetitive sequences. This analysis yielded 615 unique sequences. These were used in turn as a reference set for RNA-seq. There were four sequences that met the criterion of FDR -value <0.01. Three were transcribed at 5- to 40-fold higher levels in LPS-treated deermice

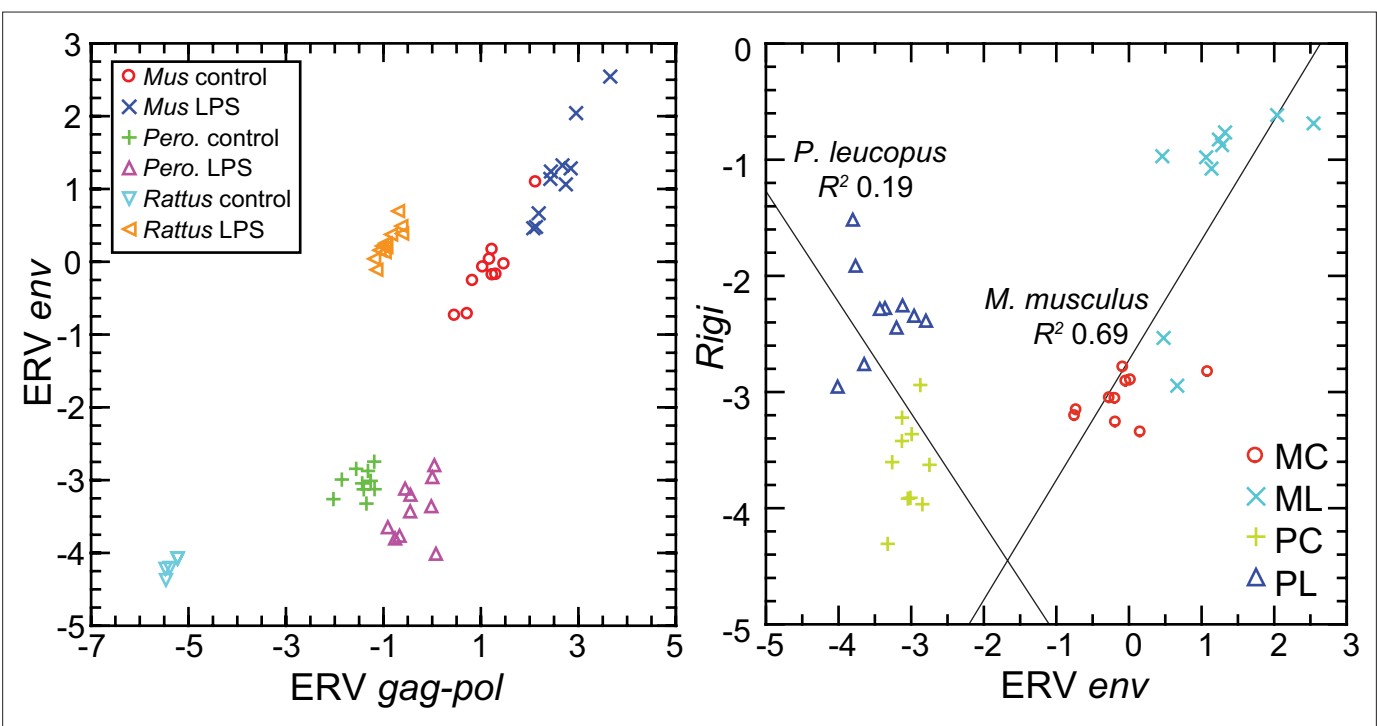

**Figure 12.** Scatter plots of endogenous retrovirus (ERV) *env* and *gag-pol* transcripts (left) and association of ERV *env* with *Rigi* transcription (right) in the blood of *Peromyscus leucopus* (Pero.; P), *M musculus* (Mus; M), or *R. norvegicus* (Rattus) with (L) or without (control; C) treatment with LPS. In right panel, the linear regression curve and coefficients of determination ($R^2$) for *P. leucopus* and *M. musculus* are shown. Values for analysis are in *Source data 1*; box plots for ERV *env* and ERV *gag-pol* transcripts are provided in *Figure 12—figure supplement 1*.

The online version of this article includes the following figure supplement(s) for figure 12:

**Figure supplement 1.** Box plots of log-transformed normalized transcripts in blood of an envelope protein protein gene (*env*) and an *gag-pol* gene of endogenous retroviruses (ERV) of *Peromyscus leucopus* (P), *Mus musculus* (M), or *Rattus norvegicus* (R) that have been treated with LPS (L) or were saline-alone controls (C).

than in controls. But all three, as well as the fourth, a down-regulated DEG, were ERV relics with truncations, frame shifts, and in-frame stop codons. These were assessed as non-coding RNAs and not further pursued in this study.

To represent *P. leucopus* in a targeted RNA-seq comparison with mice and rats, we settled on the above-referenced *env* and *gag-pol* coding sequences in blood mRNA. Representing *M. musculus* were ERV *env* transcript XM_036160206 and *gag-pol* transcript XM_036154935. For rats, we chose *env* and *gag-pol* transcripts that were second and third ranked DEGs in the genome-wide RNA-seq as noted above. Because of length differences for the coding sequences, the unit used for cross-species analysis was reads per kilobase before normalization for *Ptprc* transcription.

The left panel of *Figure 12* shows the striking transcriptional fold-change in LPS-treated rats of these *env* and *gag-pol* transcripts over controls. Of lesser magnitude but no less significant was the fold-change observed *M. musculus* for both *env* and *gag-pol* sequences. In both mice and rats, *env* and *gag-pol* read values were highly correlated across conditions. In contrast, in *P. leucopus* the magnitudes of fold-change upwards for *gag-pol* was less than in mice or rats, and transcription of the *env* sequence was actually lower in LPS-treated animals than in controls. While there was a tight association between *env* and *Rigi* transcription in the *M. musculus*, this was not observed in *P. leucopus*. *Rigi* transcription was moderately higher at the time that the *env*'s transcription was lower in the LPS group.

### *Borrelia hermsii* infection of *P. leucopus*

The phenomena reported so far were consequences of exposures to a particular PAMP—bacterial lipopolysaccharide with its hallmark lipid A moiety--recognized by a particular PRR, TLR4. While the focus was primarily on events downstream from that initial signaling, we asked in a concluding study whether the profile observed in *P. leucopus* applied in circumstances when the PAMP or PAMPs did not include LPS. This question is germane, given *P. leucopus'* role as a natural host for *B. burgdorferi*. This organism and other members of the spirochete family *Borreliaceae* do not have LPS (*Barbour, 2018*; *Takayama et al., 1987*), but they have abundant lipoproteins, which are agonists for TLR2 in a heterodimer with TLR1 (*Salazar et al., 2009*). *B. burgdorferi* is transiently blood-borne at low densities in *P. leucopus*, but in its life cycle *B. burgdorferi* is mainly tissue-associated in vertebrate hosts (*Barbour et al., 2009*). We previously observed that the blood of *B. burgdorferi*-infected *P. leucopus* manifested few DEGs in comparison to skin (*Long et al., 2019*). More comparable to the LPS experimental model is infection of *P. leucopus* with a relapsing fever *Borrelia* species, which commonly achieve high densities in the blood. *P. leucopus* is a reservoir for *Borrelia miyamotoi*, which causes hard tick-borne relapsing fever (*Barbour et al., 2009*), and the related *P. maniculatus* is a natural host for the soft tick-borne relapsing fever agent *B. hermsii* (*Johnson et al., 2016*).

Accordingly, we used blood RNA-seq reads, which were taken from a prior study of *B. hermsii* infection of *P. leucopus* (*Balderrama-Gutierrez et al., 2021*), for targeted analysis with the same reference set employed for the LPS analyses (*Table 5*; *Table 5—source data 1*). The blood samples were taken from infected and uninfected animals on day 5, when bacteremia was at its peak, as documented by microscopy of the blood, qPCR of the spleen, and transcripts of a *B. hermsii* plasmid in the RNA extracts of the blood. As expected for *B. hermsii* infection (*Crowder et al., 2016*), the spleen was enlarged in infected animals.

Similarities in the profiles for the LPS-treated and *B. hermsii*-infected deermice were as follows: (1) low levels of transcription of *Nos2* and *Ifng* that contrasted with the high levels for *Arg1* and *Il1b* expression in the same animals, (2) maintenance of the *Akt1/Akt2* ratio >1.0 under both conditions, (3) reduction of the *Cd14/Fcgr3* ratio, (4) decreased transcription of *Cd69*, and (5) stable, low transcription of ERV env and *gag-pol* loci with only marginal increases in transcription of ISGs and RIG-I-like receptors. Other equivalences under the two experimental conditions included increases in expression of genes for superoxide dismutase 2, low-affinity Fc gamma receptors, and secretory leukocyte peptidase inhibitor. Thus, the responses that distinguish deermice are not confined to the singular case of LPS as the elicitor.

**Table 5.** Targeted RNA-seq of *Peromyscus leucopus* with and without *Borrelia hermsii* infection.

| Variable | Uninfected (n=3) mean (95% CI)* | Infected (n=4) mean (95% CI)* | Fold change | FDR p value |
|---|---|---|---|---|
| *B. hermsii* qPCR of spleen | . | 13615 (1882–98,476) | . | . |
| *B. hermsii* reads blood† | . | 3487 (743–16,362) | . | . |
| % spleen/body mass | 0.15 (0.12–0.19) | 0.36 (0.26–0.51) | 2.4 | 1E-02 |
| Targeted RNA-seq | | | | |
| *Akt1* | 184 (135–253) | 347 (191–630) | 1.88 | 3E-01 |
| *Akt2* | 92.9 (72.4–119) | 141 (89.4–222) | 1.52 | 3E-01 |
| *Arg1* | 247 (96.9–630) | 1375 (848–2230) | 5.57 | 4E-02 |
| *Cd14* | 308 (119–799) | 598 (357–1002) | 1.94 | 3E-01 |
| *Cd177* | 2.11 (0.71–6.27) | 35.9 (11.9–108) | 17.0 | 4E-02 |
| *Cd69* | 133 (114–155) | 65 (33.6–128) | 0.49 | 2E-02 |
| *Cxcl10* | 0.33 (0.14–0.76) | 4.85 (2.74–8.59) | 14.7 | 2E-02 |
| ERV *env* | 26.8 (22.4–32.1) | 29.2 (14.8–57.8) | 1.09 | 9E-01 |
| ERV *gag-pol* | 1853 (1351–2542) | 1578 (820–3037) | 0.85 | 8E-01 |
| *Fcgr2a* | 44.8 (24.0–83.6) | 715 (303–1689) | 16.0 | 2E-02 |
| *Fcgr2b* | 47.3 (30.2–74.3) | 578 (167–2000) | 12.2 | 5E-02 |
| *Fcgr3* | 30.6 (18.6–50.4) | 392 (171–897) | 12.8 | 2E-02 |
| *Gapdh* | 1985 (1142–3447) | 5366 (2383–12,081) | 2.70 | 2E-01 |
| *Gbp4* | 126 (82.3–193) | 289 (130–644) | 2.30 | 3E-01 |
| *Ifit1* | 223 (107–465) | 604 (303–1203) | 2.71 | 2E-01 |
| *Ifng* | 0.58 (0.09–3.92) | 2.28 (0.86–6.06) | 3.94 | 3E-01 |
| *Il10* | 0.25 (0.07–0.87) | 1.49 (0.31–7.28) | 5.94 | 3E-01 |
| *Il12* | 0.43 (0.23–0.81) | 1.13 (0.45–2.88) | 2.63 | 3E-01 |
| *Il1b* | 477 (174–1308) | 2828 (1325–6034) | 5.93 | 7E-02 |
| *Irf7* | 93.3 (13.4–65) | 626 (196–1998) | 6.71 | 2E-01 |
| *Isg15* | 302 (30.1–3030) | 1922 (623–5934) | 6.36 | 3E-01 |
| *Itgam* | 72.2 (44.5–117) | 322 (211–492) | 4.45 | 2E-02 |
| *Mmp8* | 7.1 (2.74–18.6) | 537 (148–1952) | 75.2 | 2E-02 |
| *Mx2* | 152 (48.6–476) | 167 (48.0–582) | 1.10 | 9E-01 |
| *Nos2* | 0.16 (0.08–0.30) | 0.32 (0.13–0.80) | 2.01 | 4E-01 |
| *Oas1* | 51.3 (6.75–390) | 159 (39.2–643) | 3.10 | 5E-01 |
| *Rigi* (*Ddx58*) | 38.7 (18.9–79.2) | 55.7 (33.9–91.5) | 1.44 | 5E-01 |
| *S100a9* | 1739 (657–4596) | 18430 (6546–51,883) | 10.6 | 5E-02 |
| *Saa3* | 0.49 (0.13–1.87) | 212 (25.9–1733) | 431 | 2E-02 |
| *Slpi* | 0.41 (0.18–0.95) | 166 (49.0–566) | 401 | 1E-02 |
| *Sod2* | 104 (51.4–211) | 2011 (804–5028) | 19.3 | 2E-02 |
| *Tlr2* | 83.2 (59.7–116) | 371 (234–587) | 4.46 | 2E-02 |
| *Tlr4* | 44.8 (26.1–77.0) | 256 (138–474) | 5.71 | 3E-02 |
| Ratios | | | | |

*Table 5 continued on next page*

Table 5 continued

| Variable | Uninfected (n=3) mean (95% CI)* | Infected (n=4) mean (95% CI)* | Fold change | FDR p value |
|---|---|---|---|---|
| Akt1/Akt2 | 2.0 (1.7–2.3) | 2.5 (2.1–2.9) | 1.25 | 1E-01 |
| Cd14/Fcgr3 | 11.3 (4.8–17.9) | 1.7 (0.72–2.8) | 0.15 | 2E-02 |
| IL12/IL10 | 2.3 (0.00–5.0) | 2.8 (0.0–7.6) | 1.21 | 9E-01 |
| Nos2/Arg1 | 0.001 (0.0–0.002) | 0.0001 (0.0–0.0004) | 0.28 | 2E-01 |

*For targeted RNA-seq it is mean unique reads for given gene normalized for reads for Ptprc (Cd45) gene for a sample. The 95% confidence intervals (CI) are asymmetric. Actual [gene]/Ptprc ratios are x 10–3.
†† Normalized PE150 reads mapping to cp6.5 plasmid of *B. hermsii*.

The online version of this article includes the following source data for table 5:

**Source data 1.** Targeted RNA-seq of blood with normalization by Ptprc of P. leucopus with and without infection by Borrelia hermsii.

## Discussion
### Study limitations

The approach was forward and unbiased, looking for differences between species broadly across their transcriptomes. The findings lead to hypotheses, but reverse genetics in service of that testing was not applied here. In selective cases we could point to supporting evidence in the literature on *M. musculus* and the phenotypes of relevant gene knockouts, but there are no such resources for *Peromyscus* as yet. The resource constraint also applies to the availability of antibodies for use with *Peromyscus* for immunoassays for specific proteins, for example interferon-gamma, in serum, or for cell markers, for example CD69, for flow cytometry of white blood cells.

While a strength of the study was use of an outbred population of *M. musculus* to approximate the genetic diversity of the *P. leucopus* in the study, this meant that some genes of potential relevance might have gone undetected, that is from type II error. The variances for a sample of genetically diverse outbred animals, like the LL stock of *P. leucopus* (***Long et al., 2019***; ***Long et al., 2022***), would be expected to be greater than for the same sized sample of inbred animals. For some traits, especially ones that are complex or under balancing selection, even sample sizes of 10 in each group may not have provided sufficient power for discrimination between deermice and mice. For the same reason differences between sexes of a species in their responses might have been undetected. The interpretations applied to mixed-sex groups of deermice and mice. Expression strongly associated with female or male sex could have yielded an average fold change for the whole group that fell below the screen's threshold.

The parameters for the experiment of LPS dose, the route, and duration of experiment each might have had different values under another design. Those particular choices were based on past studies of deermice and mice (***Balderrama-Gutierrez et al., 2021***; ***Langeroudi et al., 2014***). In another experiment, we found that with doses twice or half those given the deermice the responses by rats to the different doses were indistinguishable by hematology, cytokine assays, and RNA-seq. Thus, there seems to be some latitude in the dose and still achieving replication. We obtained similar results for *P. leucopus* when we looked at a replicate of the experiment with the same conditions (***Balderrama-Gutierrez et al., 2021***), or when the dose was lower and duration lengthened to 12 hr (this study). The analysis here of the *B. hermsii* infection experiment also indicated that the phenomenon observed in *P. leucopus* was not limited to a TLR4 agonist.

While the rodents in these experiments were housed in the same facility and ate the same diet, we cannot exclude inherent differences in gastrointestinal microbiota between species and individual outbred animals as co-variables for the experimental outcomes. We reported differences between the LL stock *P. leucopus* and BALB/c *M. musculus* of the same age and diet in their microbiomes by metagenomic analysis and microbiologic means (***Milovic et al., 2020***). This included a commensal *Tritrichomonas* sp. in *P. leucopus* but not in the *M. musculus* in the study. The presence of these protozoa affects innate and adaptive immune responses in the gastrointestinal tract (***Chiaranunt et al., 2022***;

*Escalante et al., 2016*), but it is not clear whether there are systemic consequences of colonization by this flagellate.

## LPS, ERVs, and interferons

The results confirm previous reports of heightened transcription of ERV sequences in mice or mouse cells after exposure to LPS (*Hara et al., 1981*; *Jongstra and Moroni, 1981*; *Stoye and Moroni, 1983*). Here we add the example of the rat. The LPS was administered in solution and not by means of membrane vesicles. The sensing PRR presumably was surface-displayed, membrane-anchored TLR4 (*Mazgaeen and Gurung, 2020*). It follows that a second, indirect of LPS on the mouse is through its provocation of increased ERV transcription intracellularly. ERV-origin RNA, cDNA and/or protein would then be recognized by a cytoplasmic PRR. RIG-I was one associated with ERV transcription in this study. Kong et al. reported that LPS stimulated expression of *Rigi* in a mouse macrophages but did not investigate ERVs for an intermediary function in this phenomenon (*Kong et al., 2009*). As was demonstrated for LINE type retrotransposons in human fibroblasts, intracellular PRR signaling can trigger a type 1 interferon response (*De Cecco et al., 2019*). The combination of these two signaling events, that is one through surface TLR4 by LPS itself and another through intracellular PPR(s) by to-be-defined ERV products, manifested in mice and rats as a response profile that had features of both a response to a virus with type 1 interferon and ISGs and a response to a bacterial PAMP like LPS with acute phase reactants such as calprotectin and serum amyloid.

This or a similar phenomenon has been observed under other circumstances. In humans, there was heightened transcription of retrotransposons in patients with septic shock (*Mommert et al., 2020*), as well as in peripheral blood mononuclear cells from human subjects experimentally injected with LPS (*Pisano et al., 2020*). Bacteria like *Staphylococcus epidermidis* that express TLR2 agonists, such as lipoteichoic acid, promoted expression of ERVs, which in turn modulated host immune responses (*Lima-Junior et al., 2021*). A synthetic analog of a *B. burgdorferi* lipoprotein activated human mono-cytic cells and promoted replication of the latent HIV virus in cells that were persistently infected (*Norgard et al., 1996*).

*P. leucopus* does not fit well with this model. Instead of the prominent interferon-gamma response observed in mice and rats, there were prominent responses of interleukin-1 beta and genes associated with neutrophil activation. Instead of the much heightened expression of ISGs, like *Mx2* and I*sg15*, in mice treated with LPS, the deermice under the same condition had a more subdued ISG transcription profile. Instead of increased expression of ERV Env protein sequences in blood of mice and rats treated with LPS, there was decreased transcription of the homologous ERV *env* in like-treated *P. leucopus*.

This suppression in the deermice may be attributable to defensive adaptations of *Peromyscus* to repeated invasions of endogenous retroviruses, as Gozashti et al. has proposed for *P. maniculatus* (*Gozashti et al., 2023*). This includes expanding the repertoire of silencing mechanisms, such as Kruppel-associated box (KRAB) domain-containing zinc finger proteins (*Yang et al., 2017*). Like *P. maniculatus*, *P. leucopus* has an abundance of Long Terminal Repeat retrotransposons, several named for their endogenous retrovirus heritages (*Long et al., 2019*). Our initial analysis of the *P. leucopus* genome reported a depletion of KRAB domains compared to Muridae (*Long et al., 2019*). But a subsequent annotation round identified several genes for KRAB domain zinc finger proteins in *P. leucopus*, including Zfp809 (XP_006982432), which initiates ERV silencing (*Wolf et al., 2015*), and Zfp997 (XP_037067826), which suppresses ERV expression (*Treger et al., 2019*). Another possible adaptation in *P. leucopus* is the higher baseline expression of some ISGs as noted here (*Figure 11*; *Figure 11—figure supplement 1*).

Reducing differences between *P. leucopus* and murids *M. musculus* and *R. norvegicus* to a single attribute, such as the inactivation of *Fcgr1* in *P. leucopus* (*Barbour et al., 2023*), may be fruitless. But the feature that may best distinguish the deermouse from the mouse and rat is its predominantly anti-inflammatory quality. This characteristic likely has a complex, polygenic basis, with environmental (including microbiota) and epigenetic influences. An individual's placement is on a spectrum or, more likely, a landscape rather than in one or another binary or Mendelian category.

One argument against a purely anti-inflammatory characterization is the greater neutrophil numbers and activity in *P. leucopus* compared to *M. musculus* in the LPS experiment. The neutrophil activation, migration, and phagocytosis would be appropriate early defenses against a pyogenic pathogen. But

if not contained, they bring local and systemic risks for the host. This damage would not likely be from nitric oxide and reactive nitrogen species, given the minimal *Nos2* transcription. But deermice showed heightened expression of genes for proteases, such as *Mmp8*, enzymes for reactive oxygen species, such as NADPH oxidase 1 (*Nox1*), and facilitators of neutrophil extracellular traps, such as PAD4 (*Padi4*) (*Supplementary file 1* ). We had previously identified possible mitigators, such as secretory leuckocyte peptidase inhibitor and superoxide dismutase 2 (*Balderrama-Gutierrez et al., 2021*). These findings were replicated here. The topic of neutrophil activation and these and other possible counters is considered in more detail elsewhere.

## An anti-inflammatory disposition but at what cost?

An assignment of infection tolerance to a host and pathogen pairing assumes sufficient immunity against the microbe to keep it in check if elimination fails. *P. leucopus* and *P. maniculatus*, are in this sense 'immunocompetent' with respect to the microbes they host and with which they may share a long history (*Hoen et al., 2009*). Yet, has this balance of resistance and tolerance for certain host-associated microbes been achieved in a trade-off that entails vulnerabilities to other types of agents?

The selection of LPS as the experimental model was meant to cover this contingency, at least for the common denominator of acute inflammation many types of infections elicit. But LPS studies revealed potential weaknesses that some pathogens might exploit. One of these is the low expression of inducible nitric oxide. Although *Nos2* gene knockouts in *M. musculus* had lower LPS-induced mortality than their wild-type counterparts, the mutants were more susceptible to the protozoan *Leishmania major* and the facultative intracellular bacterium *Listeria monocytogenes* (*MacMicking et al., 1995*; *Wei et al., 1995*). While there are no known studies of either of these pathogens in *P. leucopus*, the related species *P. yucatanicus* is the main reservoir for *Leishmania mexicana* in Mexico (*Chable-Santos et al., 1995*). Compared with *M. musculus*, which suffer a high fatality rate from experimental infections with *L. mexicana*, *P. yucatanicus* infections are commonly asymptomatic (*Loría-Cervera et al., 2018*).

Given the restrained interferon and ISG response shown by *P. leucopus*, another plausible vulnerability would be viral infections. But other studies suggest that neither RNA nor DNA viruses pose an inordinately high risk for *Peromyscus*. Both tolerance of and resistance to the tickborne encephalitis flavivirus Powassan virus by *P. leucopus* were demonstrated in an experimental model in which mice, by contrast, were severely affected *Mlera et al., 2017*. *P. maniculatus* has been successfully infected with the SARS-CoV-2 virus by the respiratory route, but the infected animals displayed only mild pathology, manifested little if any disability, and recovered within a few days *Fagre et al., 2021*; *Griffin et al., 2021*. Among natural populations and in the laboratory, *P. maniculatus* is noted for its tolerance of hantavirus, which commonly is fatal for infected humans *Botten et al., 2000*; *Childs et al., 1994*. *P. maniculatus* was permissive of infection with monkeypox virus, but the infection was mild and transient *Deschambault et al., 2023*.

A distinguishing *P. leucopus* characteristic, which was not expressly examined here, is its aforementioned two- to threefold greater life span than that of *M. musculus*. While deermice may not be in the same longevity league as the naked mole-rat (*Heterocephalus glaber*), which can live for over 30 years *Oka et al., 2023*, some features of naked mole-rat immunology are intriguingly similar to what we have observed for *P. leucopus*. These include macrophages and blood myeloid cells with low to absent transcription of *Nos2* or production of nitric oxide in response to LPS, even in the presence of added interferon-gamma *Gorshkova et al., 2023*. Like *P. leucopus* and in distinction to *M. musculus*, naked mole-rats showed an increase in the proportion of neutrophils in the blood 4 hr after intraperitoneal injection of LPS *Hilton et al., 2019*. In another comparative study, the hematopoietic stem and progenitor cells of these rodents had a lower type 1 interferon response than mice to a TLR3 agonist *Emmrich et al., 2022*.

In summary, if there is a vulnerability that *Peromyscus* accepts in return for relief from inflammation (and perhaps a longer life), it has not been identified yet. However, potential threats and stressors are many, and the number assessed either in the field or laboratory has been limited to date.

## Implications for Lyme disease and other zoonoses

Our studies of *P. leucopus* began with a natural population and documented a>80% prevalence of infection and high incidence of re-infections by *B. burgdorferi* in the area's white-footed deermouse, the most abundant mammal there (*Bunikis et al., 2004*). This was a Lyme disease endemic area

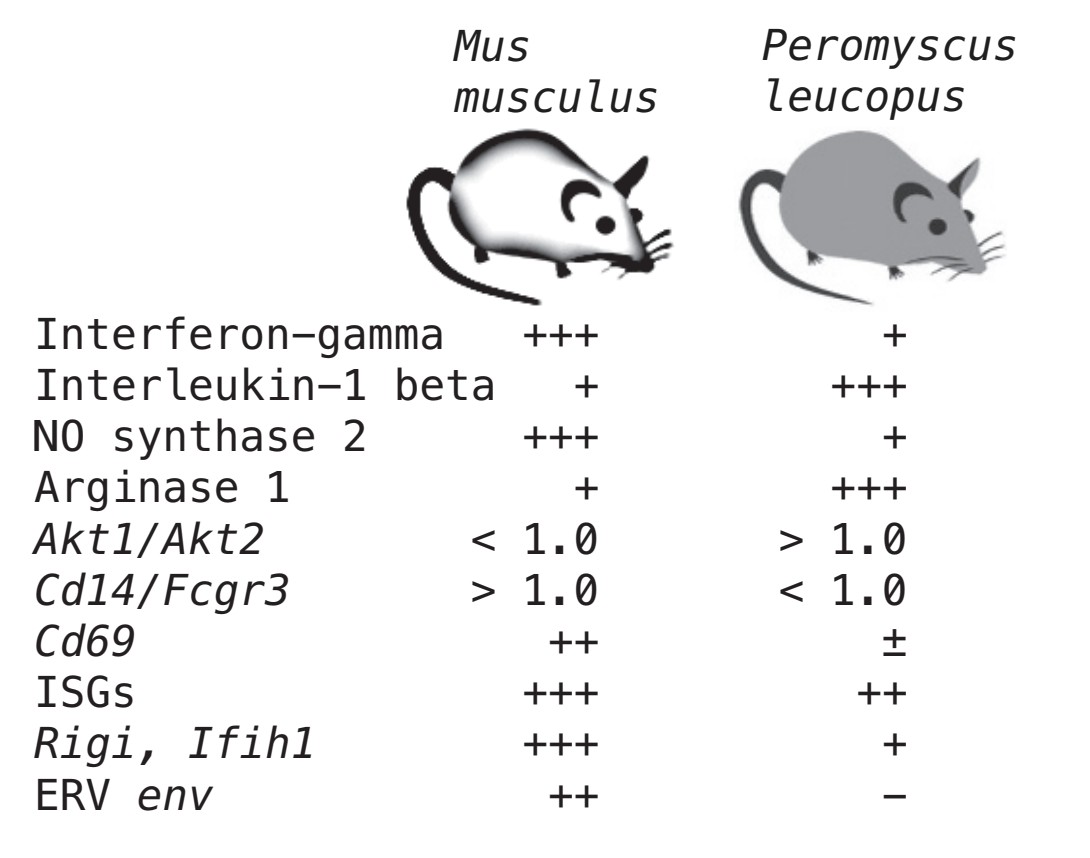

**Figure 13.** Summary of distinguishing features of transcriptional responses in the blood between *Peromyscus leucopus* and *Mus musculus* 4 hr after treatment with LPS. There is semi-quantitative representation of relative transcription of selected coding sequences or ratios of transcription for selected pairs of genes in the blood.

(*Brinkerhoff et al., 2012*), where residents frequently presented for medical care for a variety of clinical manifestations, from mild to serious, of *B. burgdorferi* infection (*Steere et al., 1986*). Subclinical infections in humans occur, but most of those who become infected have a definable illness (*Steere et al., 1998*). The localized or systemic presence of the microbe is a necessary condition for Lyme disease, but the majority of the symptoms and signs are attributable to inflammation elicited by the organism's presence and not from virulence properties per se or the hijacking of host cells (*Coburn et al., 2022*). Since humans are transmission dead-ends for *B. burgdorferi* and many other zoonotic agents in their life cycles, it is not surprising that human infections are generally more debilitating if not fatal than what adapted natural hosts experience.

It is in the space between the asymptomatic natural host and symptomatic inadvertent host where there may be insights with basic and translational application. With this goal, we consider the ways the results inform studies of the pathogenesis of Lyme disease, where 'disease' includes lingering disorders akin to 'long Covid' (*Nathan, 2022*), and where 'pathogenesis' includes both microbial and host contributions. Plausibly germane deermouse-mouse differences identified in our studies are summarized in *Figure 13*. Two are highlighted here.

The first is macrophage polarization (*Murray, 2017*). By the criteria summarized above, the response to LPS by *P. leucopus* is consistent with the alternatively-activated or M2 type, rather than the expected classical or M1 type. But it was not only LPS-treated deermice that had this attribute, the blood of untreated animals also displayed M2 type polarization features. This included a comparatively high *Arg1* expression level and a *Akt1/Akt2* transcription ratio of more than 1 at baseline. This suggests that studies of other mammals, including humans, need not administer LPS or other TLR agonist to assess disposition toward M1 or M2-type polarization. This reading could serve as a

prognostic indicator of the inflammatory response to infection with *B. burgdorferi* or other pathogen and the long-term outcome.

The second difference we highlight is the activation of ERV transcription that was prominent in the LPS-treated mice and rats but not in similarly-treated deermice. A paradoxical enlistment of antiviral defenses, including type 1 and type 2 interferons, for an infection with an extracellular bacterium, like *B. burgdorferi*, may bring about more harm than benefit, especially if the resultant inflammation persists after antibiotic therapy. There are various ways to assess ERV activation in the blood, including assays for RNA, protein, and reverse transcriptase activity. A xenotropic MLV-related retrovirus has been discounted as a cause of chronic fatigue syndrome (*McClure and Kaye, 2010*). However, production of whole virions need not occur for there to be PRR signaling in response to cytoplasmic Env protein, single stranded RNA, or cDNA (*Russ and Iordanskiy, 2023*).

# Methods

## Key resources table

| Reagent type (species) or resource | Designation | Source or reference | Identifiers | Additional information |
|---|---|---|---|---|
| Strain, strain background (*Peromyscus leucopus*) | Outbred LL stock; adults of both sexes | Peromyscus Genetic Stock Center of the University of South Carolina | | |
| Strain, strain background (*Mus musculus*) | Outbred CD-1 breed; adults of both sexes | Charles River Laboratories | Crl:CD1(ICR) IGS | |
| Strain, strain background (*Rattus norvegicus*) | Inbred Fischer F344 strain; adult females | Charles River Laboratories | F344/NHsd | |
| Strain, strain background (*Borrelia hermsii*) | Genomic group II, strain MTW | *Balderrama-Gutierrez et al., 2021* (reference 3) | | Provided by Tom Schwan, Rocky Mountain Laboratories |
| Sequence-based reagent | Arg1_F for *P. leucopus* | This paper | PCR primer | TCCGCTGACAACCAACTCTG |
| Sequence-based reagent | Arg1_R for *P. leucopus* | This paper | PCR primer | GACAGGTGTGCCAGTAGATG |
| Sequence-based reagent | Arg1_F for *M. musculus* | This paper | PCR primer | TGTGAAGAACCCACGGTCTG |
| Sequence-based reagent | Arg1_R for *M. musculus* | This paper | PCR primer | ACGTCTCGCAAGCCAATGTA |
| Sequence-based reagent | Nos2_F for *P. leucopus* and *M. musculus* | *Balderrama-Gutierrez et al., 2021* (reference 3) | PCR primer | GACTGGATTTGGCTGGTCCC |
| Sequence-based reagent | Nos2_R for *P. leucopus* and *M. musculus* | *Balderrama-Gutierrez et al., 2021* (reference 3) | PCR primer | GAACACCACTTTCACCAAGAC |
| Sequence-based reagent | Gapdh_F for *P. leucopus* and *M. musculus* | *Balderrama-Gutierrez et al., 2021* (reference 3) | PCR primer | TCACCACCATGGAGAAGGC |
| Sequence-based reagent | Gapdh_R for *P. leucopus* and *M. musculus* | *Balderrama-Gutierrez et al., 2021* (reference 3) | PCR primer | GCTAAGCAGTTGGTGGTGCA |
| Commercial assay or kit | Invitrogen Mouse RiboPure-Blood RNA Isolation Kit | Invitrogen | AM1951 | |
| Commercial assay or kit | TruSeq Stranded mRNA kit for cDNA | Illumina | 20020594 | |
| Commercial assay or kit | Power Sybr Green RNA-to-Ct 1-Step Kit for RT-qPCR | Applied Biosystems | ThermoFisher 4389986 | |
| Chemical compound, drug | Lipopolysaccharide, *Escherichia coli* O111:B4, ion-exchange chromatography purified | Sigma-Aldrich | L3024 | |
| Chemical compound, drug | Lipopolysaccharide, *Escherichia coli* O111:B4, cell culture grade" | Sigma-Aldrich | L4391 | |
| Chemical compound, drug | 0.9% sodium chloride sterile-filtered, endotoxin-tested | Sigma-Aldrich | S8776 | |
| Software, algorithm | FastQC, version 0.12.0 | Babraham Bioinformatics | | https://www.bioinformatics.babraham.ac.uk/projects/fastqc/ |
| Software, algorithm | Trimmomatic, version 0.40 | USADELLAB.org; *Usadel and Bolger, 2023* | | https://github.com/usadellab/Trimmomatic |
| Software, algorithm | CLC Genomics Workbench, version 23.1 | Qiagen | | |
| Software, algorithm | EnrichR (Enrichment of Gene Ontology) | Metascape | | https://metascape.org |
| Software, algorithm | SYSTAT, version 13.1 | Systat Software, Inc | | |
| Software, algorithm | False Discovery Rate Online Calculator | Carbocation Corporation | | https://tools.carbocation.com/FDR |

## Animals

The study was carried out in accordance with the *Guide for the Care and Use of Laboratory Animals: Eighth Edition* of the National Academy of Sciences, and according to ARRIVE Guidelines (arrive-guidelines.org). The protocols AUP-18–020 and AUP-21–007 were approved by the Institutional Animal Care and Use Committee of the University of California Irvine.

*Peromyscus leucopus*, here also referred to as 'deermice', were of the outbred LL stock, which originated with 38 animals captured near Linville, NC, and thereafter comprised a closed colony without sib-sib matings at the *Peromyscus* Genetic Stock Center at the University of South Carolina (*Joyner et al., 1998*). LL stock animals for this study were bred and raised at the vivarium of University of California Irvine, an AAALAC approved facility. Outbred *Mus musculus* breed CD-1 (Crl:CD1(ICR) IGS), and here also referred to as 'mice', were obtained from Charles River Laboratories. Fischer F344 strain inbred *Rattus norvegicus* (F344/NHsd), here also referred to as "rats", were obtained from Charles River Laboratories. Facility acclimatization was for at least 1 week before study.

For the combined *P. leucopus-M. musculus* experiment, the 20 *P. leucopus* were of a mean (95% confidence interval) 158 (156-159) days of age and had a mean 21 (19-22) g body mass. The 20 *M. musculus* were all 149 days of age and had a mean body mass of 47 (43-50) g. The ratio of average male to average female body mass was 1.04 for *P. leucopus* and 1.03 for *M. musculus*. The six female *P. leucopus* for the 12 hr duration experiment were of a mean 401 (266-535) days of age and mean body mass of 20 (17-23) g. The 16 adult 10- to 12-week-old female *R. norvegicus* had a mean 139 (137-141) g body mass. The seven male *P. leucopus* for the infection study were of a mean 107 (80-134) days and mean body mass of 21 (18-24) g.

Animals were housed in Techniplast-ventilated cages in vivarium rooms with a 16 h-8 h light-dark cycle, an ambient temperature of 22 °C, and on ad libitum water and a diet of 2020 X Teklad global soy protein-free extruded rodent chow with 6% fat content (Envigo, Placentia, CA).

For all injections, the rodents were anesthetized with inhaled isoflurane. The rodents were euthanized by carbon dioxide overdose and intracardiac exsanguination at the termination of the experiment. No animals died or became moribund before the 4 hr or 12 hr termination time points in the LPS experiments or before the 5 d termination point of infection study.

## LPS and infection model experiments

For the *P. leucopus* and *M. musculus* combined experiment, sample sizes replicated the specifications of the previous study, in which there were 20 *P. leucopus* and 20 *M. musculus*, equally divided between females and males and equally allotted between conditions (*Balderrama-Gutierrez et al., 2021*). The treatments were administered in the morning of a single day. At 15 min intervals and alternating between species, sex, and treatments, animals were intraperitoneally (ip) injected 50 μl volumes of either ion-exchange chromatography-purified *Escherichia coli* O111:B4 LPS (Sigma-Aldrich L3024) in a dose of 10 μg per g body mass or the diluent alone: sterile-filtered, endotoxin-tested, 0.9% sodium chloride (Sigma-Aldrich). The animals were visually monitored in separate cages continuously for the duration of the experiment. We recorded whether there was reduced activity by criterion of huddling with little or movement for >5 min, ruffled fur or piloerection, or rapid respiration rate or tachypnea. At 4.0 hr time after injection animals were euthanized as described above, and sterile dissection was carried out immediately.

Lower dose and longer duration experiment. In an experiment with six *P. leucopus*, the animals were administered the same single dose of LPS but at 1.0 μg/g and the same control solution. The animals were euthanized 12 hr after the injection the following day.

Rat LPS experiment. The same experimental design was used for the rats as for the combined deermice-mice experiment, with the exception that the formulation of the *E. coli* O111:B4 LPS was 'cell culture grade' (Sigma-Aldrich L4391), and the groups were sterile saline alone (n=5), 5 μg LPS per g body mass (n=6), or 20 μg LPS per g (n=5).

## Experimental infection

The infection of a group of *P. leucopus* LL stock with the relapsing fever agent *B. hermsii* and the processing of blood and tissues for RNA extraction 5 days into the infection were described previously (*Balderrama-Gutierrez et al., 2021*). In brief, animals were infected intraperitoneally on day 0 with either phosphate-buffered saline alone or $10^3$ cells of *B. hermsii* MTW, a strain that is infectious for

*Peromyscus* species (*Johnson et al., 2016*). Bacteremia was confirmed by microscopy on day 4, and the animals were euthanized on day 5. For that prior study the RNA-seq analysis was limited to the genome-wide transcript reference set. For the present study we used the original fastq format files for targeted RNA-seq as described below.

## Hematology and plasma analyte assays

For the combined *P. leucopus-M. musculus* experiment, automated complete blood counts with differentials were performed at Antech Diagnostics, Fountain Valley, CA on a Siemens ADVIA 2120i with Autoslide hematology instrument with manual review of blood smears by a veterinary pathologist. For the 12 hr duration *P. leucopus* experiment, hematologic parameters were analyzed on an ABCVet Hemalyzer automated cell counter instrument at U.C. Irvine. For the rat experiment, complete blood counts with differentials were performed at the Comparative Pathology Laboratory of the University of California Davis. Multiplex bead-based cytokine protein assay of the plasma of the rats was performed at Charles River Laboratories using selected options of the Millipore MILLIPLEX MAP rat cytokine/chemokine panel.

## RNA extraction of blood

After the chest cavity was exposed, cardiac puncture was performed through a 25 gauge needle into a sterile 1 ml polypropylene syringe. After the needle was removed, the blood was expelled into Becton-Dickinson K2E Microtainer Tubes, which contained potassium EDTA. Anticoagulated blood was split into a sample that was placed on ice for same-day delivery to the veterinary hematology laboratory and a sample intended for RNA extraction which was transferred to an Invitrogen RiboPure tube with DNA/RNA Later and this suspension was stored at –20 °C. RNA was isolated using the Invitrogen Mouse RiboPure-Blood RNA Isolation Kit. RNA concentration was determined on a NanoDrop microvolume spectrophotometer (ThermoFisher) and quality was assessed on an Agilent Bioanalyzer 2100.

## RNA-seq of blood

The chosen sample sizes and coverage for the bulk RNA-seq were based on empirical data from the prior study (*Balderrama-Gutierrez et al., 2021*), which indicated that with 10 animals per group and a two-sided two sample *t*-test we could detect with a power of ≥0.80 and at a significance level of 0.05 a ≥1.5-fold difference in transcription between groups for a given gene. We also were guided by the simulations calculations of *Hart et al., 2013*, which indicated for a biological coefficient of variation of 0.4 within a group, a minimum depth of coverage of ≥10, and a target of ≥2 x fold change that a sample size of 7–8 was sufficient for 80% power and type I error of 5%. For the *P. leucopus* and *M. musculus* samples production of cDNA libraries was with the Illumina TruSeq Stranded mRNA kit. After normalization and multiplexing, the libraries were sequenced at the University of California Irvine's Genomic High Throughput Facility on a Illumina NovaSeq 6000 instrument with paired-end chemistry and 150 cycles to achieve ~100 million reads per sample for the combined *P. leucopus-M. musculus* experiment. The same method for producing cDNA libraries was used for the *R. norvegicus* RNA and the *P. leucopus* in the infection study, but these were sequenced on a Illumina HiSeq 4000 instrument with paired-end chemistry and 100 cycles. The quality of sequencing reads was analyzed using FastQC (Babraham Bioinformatics). The reads were trimmed of low-quality reads (Phred score of <15) and adapter sequences, and corrected for poor-quality bases using Trimmomatic (*Bolger et al., 2014*).

For the combined species experiment, the mean (95% CI) number of PE150 reads per animal after trimming for quality was 1.1 (1.0–1.2) x $10^8$ for *P. leucopus* and 1.1 (1.0–1.2) x $10^8$ for *M. musculus* (p=0.91). For *P. leucopus* of this experiment, a mean of 83% of the reads mapped to the genome transcript reference set of 54,466; mean coverages for all transcripts and for the mean 62% of reference transcripts with ≥1 x coverage were 97 x and 157 x, respectively. For *M. musculus* of this experiment, a mean 91% of the reads mapped to the genome transcript reference set of 130,329; mean coverages for all transcripts and for the mean of 21% of reference transcripts with ≥1 x coverage were 103 x and 568 x, respectively. For the lower dose-longer duration experiment with *P. leucopus* the mean number of PE150 reads was 2.5 (2.3–2.6) x $10^7$. For the rat experiment, the mean number of PE100 reads was 2.4 (2.2–2.5) x $10^7$. The PE100 Illumina reads from the *B. hermsii* infection of *P. leucopus*

study (*Balderrama-Gutierrez et al., 2021*) are from BioProject PRJNA508222, which includes SRA accession numbers (SRR8283809 and SRR8283811-SRR8283816); the mean number of reads was 4.9 (4.5–5.3) x $10^7$.

Batched fastq files were subjected to analysis with CLC Genomics Workbench version 23 (Qiagen). Library size normalization was done by the TMM (trimmed mean of M values) method of *Robinson and Oshlack, 2010*. The reference genome transcript sets on GenBank were the following: GCF_004664715.2 for *P. leucopus* LL stock, GCF_000001635.27_GRCm39 for *M. musculus* C57Bl/6, and GCF_015227675.2_mRatBN7.2 for *R. norvegicus*. The settings for PE150 reads were as follows for both strands: length fraction of 0.35, similarity fraction of 0.9, and costs for mismatch, insertion, or deletion of 3. For PE100 reads, the settings were the same except for length fraction of 0.4. Principal Component Analysis was carried with the 'PCA for RNA-Seq' module of the CLC Genomics Workbench suite of programs.

For the *P. leucopus* RNA-seq analysis, there were 54,466 reference transcripts, of which 48,164 (88%) were mRNAs with protein coding sequences, and 6302 were identified as non-coding RNAs (ncRNA). Of the 48,164 coding sequences, 40,247 (84%) had matching reads for at least one of the samples. The five most highly represented *P. leucopus* coding sequences among the matched transcripts of whole blood among treated and control animals were for hemoglobin subunits alpha and beta, the calprotectin subunits S100A8 and S100A9, and ferritin heavy chain. For the *M. musculus* analysis there were available 130,329 reference transcripts: 92,486 (71%) mRNAs with protein coding sequences and 37,843 ncRNAs. Of the coding sequences, 59,239 (64%) were detectably transcribed in one or both groups by the same criterion. The five most highly represented coding sequences of mRNAs of identified genes for *M. musculus* were for hemoglobin subunits alpha and beta, amino-levulinic synthase 2, ferritin light polypeptide 1, and thymosin beta. For *R. norvegicus,* there were 99,126 reference transcripts, of which 74,742 (75%) were mRNAs. The five most highly represented coding sequences of mRNAs of identified genes for *R. norvegicus* were for hemoglobin subunits alpha and beta, beta-2 microglobulin, ferritin heavy chain, and S100A9.

## Genome-wide differential gene expression

Differential expression between experimental conditions was assessed with an assumption of a negative binomial distribution for expression level and a separate Generalized Linear Model for each (*McCarthy et al., 2012*). Fold changes in TPM (transcripts per million) were $\log_2$-transformed. The False Discovery Rate (FDR) with corrected *p* value was estimated by the method of *Benjamini and Hochberg, 1995*. To assess the limit of detection for differentially expressed genes between 10 animals treated with LPS and 10 with saline alone, we took the data for 4650 reference transcripts for which the mean TPM across 20 *P. leucopus* was >10 and randomly permuted the data to achieve another 9 sets and calculated the fold-change of sub-groups of 10 and 10 with one random group serving as the proxy of the experimental treatment and other second as the control for each of the sets. The expectation was that mean fold-change of the 9 permuted sets and the 4650 reference sequences would be ~1. The result was a mean and median of 1.08 with a 99.9% asymmetric confidence interval for the mean of 0.81–1.49. This was an indication that the choices for sample sizes were realistic for achieving detection of ≥1.5 x fold changes.

## Gene Ontology term analysis

*M. musculus* was selected as the closest reference for the *P. leucopus* data. The analysis was implemented for data for differentially expressed genes meeting the criteria of a FDR P-value ≤0.01 and fold-change of ≥1.5. The analysis was implemented with the tools of Metascape (https://metascape.org; *Zhou et al., 2019*). Functional enrichment analysis was carried out first with the hypergeometric test and FDR p-value correction (*Benjamini and Hochberg, 1995*). Then pairwise similarities between any two enriched terms were computed based on a Kappa-test score (*Cohen, 1960*). Similarity matrices were then hierarchically clustered and a 0.3 similarity threshold was applied to trim resultant trees into separate clusters. The lower the p-value, the less the likelihood the observed enrichment is due to randomness (*Zar, 1999*). The lowest p-value term represented each cluster shown in the horizontal bar graph. Besides the terms beginning with 'GO' and referring to the Gene Ontology resource (http://geneontology.org; *Ashburner et al., 2000*), others refer to Kegg Pathway database (https://

www.kegg.jp) for 'mmu.' designations, WikiPathways database (https://www.wikipathways.org) for 'WP…' designations, and Reactome database (https://reactome.org) for 'R-MMU…' designations.

## Targeted RNA-seq

RNA-seq of selected set of protein coding sequences (CDS), which are listed below, was carried out using CLC Genomics Workbench v. 23 (Qiagen). Paired-end reads were mapped with a length fraction of 0.35 for ~150 nt reads and 0.40 for ~100 nt reads, a similarity fraction of 0.9, and costs of 3 for mismatch, insertion, or deletion to the CDS of sets of corresponding orthologous mRNAs of *P. leucopus*, *M. musculus*, and *R. norvegicus*. Preliminary expression values were unique reads normalized for total reads across all the samples without adjustment for reference sequence length, as described (*Balderrama-Gutierrez et al., 2021*). Exceptions were the endogenous retrovirus coding sequences which differed in lengths between species. For within- and cross-species comparisons, we initially normalized three different ways after quality filtering and removing vector and linker sequence: for total reads for the given sample, for unique reads for 12 S ribosomal RNA for the mitochondria of nucleated cells in the blood, and for unique reads for the gene Ptprc, which encodes CD45, a marker for both granulocytes and mononuclear cells in the blood. This is described in more detail in Results. Following the recommendation of Hedges et al. we used the natural logarithm (*ln*) of ratios (*Hedges et al., 1999*).

The target CDS were as follows: *Acod1, Akt1, Akt2, Arg1, Bcl3, Camp, Ccl2, Ccl3, Ccl4, Cd14, Cd177, Cd3d, Cd4, Cd69, Cd8, Cfb, Cgas, Csf1, Csf1r, Csf2, Csf3, Csf3r, Cx3cr1, Cxcl1, Cxcl10, Cxcl2, Cxcl3, Dhx58, Fcer2, Fcgr2a, Fcgr2b, Fcgr3, Fgr, Fos, Fpr2, Gapdh, Gbp4, Glrx, Gzmb, Hif1a, Hk3, Hmox1, Ibsp, Icam, Ifih1, Ifit1, Ifng, Il10, Il12, Il18, Il1b, Il1rn, Il2ra, Il4ra, Il6, Il7r, Irf7, Isg15, Itgam, Jak1, Jak2, Jun, Lcn2, Lpo, Lrg, Lrrk2, Ltf, Mapk1, Mmp8, Mmp9, Mpo, MT-Co1, Mt2, Mtor, Mx2, Myc, Myd88, Ncf4, Nfkb1, Ngp, Nos2, Nox1, Nr3c1, Oas1, Olfm4, Padi4, Pbib, Pkm, Ptx, Ptprc, Retn, Rigi* (*Ddx58*), *S100a9, Saa3, Serpine1, Slc11a1, Slpi, Socs1, Socs3, Sod2, Stat1, Stat2, Stat4, Steap1, Sting, Tgfb, Thy1, Timp1, Tlr1, Tlr2, Tlr4, Tnf, Tnfrsf1a*, and *Tnfrsf9*. The sources for these coding sequences were the reference genome transcript sets for *P. leucopus*, *M. musculus*, and *R. norvegicus* listed above. If there were two or more isoforms of the mRNAs and the amino acid sequences differed, the default selection for the coding sequence was the first listed isoform. The lengths of the orthologous pairs of *P. leucopus* and *M. musculus* coding sequences were either identical or within 2% of the other. Fcgr1, the gene for high affinity Fc gamma receptor I or CD64, was not included in the comparison, because in *P. leucopus* it is an untranscribed pseudogene (*Barbour et al., 2023*). For the targeted RNA-seq of blood of deermice infected with *B. hermsii* (*Table 5*), the reference sequence was the cp6.5 plasmid of *B. hermsii* (NZ_CP015335).

## Quantitative PCR assays

Reverse transcriptase (RT)-qPCR assays and the corresponding primers for measurement of transcripts of genes for nitric oxide synthase 2 (*Nos2*) and glyceraldehyde 3-phosphate dehydrogenase (*Gapdh*) were those described previously (*Balderrama-Gutierrez et al., 2021*). These primers worked for *M. musculus* as well as *P. leucopus* using modified cycling conditions. For the *Arg1* transcript assays different primer sets were used for each species. The forward and reverse primer sets for the 352 bp *Arg1* product for *P. leucopus* were 5′-TCCGCTGACAACCAACTCTG and 5′-GACAGGTGTGCCAGTA GATG, respectively. The corresponding primer pairs for a 348 bp Arg1 of *M. musculus* were 5′-TGTG AAGAACCCACGGTCTG and 5′-ACGTCTCGCAAGCCAATGTA. cDNA synthesis and qPCR were achieved with a Power Sybr Green RNA-to-Ct 1-Step Kit (Applied Biosystems) in 96 MicroAmp Fast Reaction Tubes using an Applied Biosystems StepOne Plus real-time PCR instrument. The initial steps for all assays were 48 °C for 30 min and 95 °C for 10 min. For Arg1 and Nos2 assays, this was followed by 40 cycles of a 2-step PCR of, first, 95 °C for 15 s and then, second, annealing and extension at 60 °C for 1 min. The cycling conditions for Gapdh were 40 cycles of 95 °C for 15 s followed by 60 °C for 30 s. Quantitation of genome copies of *B. hermsii* in extracted DNA was carried out by probe-based qPCR as described (*Barbour et al., 2009*).

## Additional statistics

Means are presented with asymmetrical 95% confidence intervals (CI) to accommodate data that was not normally distributed. Parametric (*t* test) and non-parametric (Mann-Whitney) tests

of significance were two-tailed. Unless otherwise stated, the *t* test *p* value is given. Categorical variables were assessed by two-tailed Fisher's exact test. FDR correction of p values for multiple testing was by the Benjamini-Hochberg method (*Benjamini and Hochberg, 1995*), as implemented in CLC Genomics Workbench (see above), or False Discovery Rate Online Calculator (https://tools.carbocation.com/FDR). Discriminant Analysis, linear regression, correlation, coefficient of determination, and General Linear Model analyses were performed with SYSTAT v. 13.1 software (Systat Software, Inc). Box plots with whiskers display the minimum, first quartile, median, third quartile, and maximum.

## Acknowledgements

We thank Hanjuan Shao for technical assistance, Vanessa Cook for her participation in the *R. norvegicus* experiment, Anthony Long for bioinformatic advice and contributions, and Brianna Craver-Hoover and Gajalakshmi Ramanathan for assistance with blood cell counts performed at U.C. Irvine. The experimental studies reported here were supported by National Institutes of Health (NIH) grants AI157513 and AI136523. The services of the Genomics Research and Technology Hub were administratively supported in part by NIH Cancer Center Support Grant P30 CA-062203 and NIH shared instrumentation grants RR-025496, OD-010794, and OD-021718.

## Additional information

### Funding

| Funder | Grant reference number | Author |
| --- | --- | --- |
| National Institutes of Health | AI157513 | Alan G Barbour |
| National Institutes of Health | AI136523 | Alan G Barbour |

The funders had no role in study design, data collection and interpretation, or the decision to submit the work for publication.

### Author contributions

Ana Milovic, Conceptualization, Data curation, Formal analysis, Investigation, Methodology, Writing – review and editing; Jonathan V Duong, Resources, Data curation, Formal analysis, Investigation, Methodology, Writing – review and editing; Alan G Barbour, Conceptualization, Formal analysis, Funding acquisition, Investigation, Methodology, Writing – original draft, Project administration, Writing – review and editing

### Author ORCIDs

Alan G Barbour ⬤ https://orcid.org/0000-0002-0719-5248

### Ethics

The study was carried out in accordance with the Guide for the Care and Use of Laboratory Animals: Eighth Edition of the National Academy of Sciences, and according to ARRIVE Guidelines (arrive-guidelines.org). The protocols AUP-18-020 and AUP-21-007 were approved by the Institutional Animal Care and Use Committee of the University of California Irvine. For all injections the rodents were anesthetized with inhaled isoflurane. The rodents were euthanized by carbon dioxide overdose and intracardiac exsanguination at the termination of the experiment. No animals died or became moribund before the 4 hour or 12 h termination time points in the LPS experiments or before the 5 d termination point of infection study.

Reviewer #1 (Public Review): https://doi.org/10.7554/eLife.90135.3.sa1
Reviewer #2 (Public Review): https://doi.org/10.7554/eLife.90135.3.sa2
Author Response https://doi.org/10.7554/eLife.90135.3.sa3

## Additional files

### Supplementary files

• MDAR checklist

• Supplementary file 1. Targeted RNA-seq of blood for 115 selected genes of *Peromyscus leucopus* and *Mus musculus* 4 hours after intraperitoneal injection of lipopolysaccharide (LPS) or saline control. The list includes the 113 coding sequences listed in Methods, as well as endogenous retrovirus (ERV) *env* transcripts and ERV *gag-pol* transcripts. The values are mean unique reads with asymmetric 95% confidence intervals for a given gene transcript normalized for reads for *Ptprc* transcripts for a sample. Each species is analyzed separately with respect to LPS-treated and control animals. For cross-species comparisons the fold-change (FC) differences between control and LPS-treated animals were individually determined. For a summary representation of the differences between the two species in responses to LPS, the ratios of *P. leucopus* to *M. musculus* FC values were calculated. These are schematically displayed as a color-coded heat-map as well.

• Source data 1. Targeted RNA-seq with normalization by Ptrc transcripts of blood of P. leucopus LL stock or M. musculus CD-1 with or without treatment with LPS by individual animal.

### Data availability

Sequencing data as fastq files of Illumina reads (SRA), along with descriptions of the samples (BioSamples) they are associated with, have been deposited with NCBI under BioProjects PRJNA975149, PRJNA874306, and PRJNA973677. The Transcriptome Shotgun Assembly of the blood of Peromyscus leucopus and produced for this study has TSA accession number GKOE00000000.1.All the data generated and analyzed for this study are included in the manuscript and supporting files. Source Data files have been provided for all figures except Figure 1, the data for which are provided in Table 1.

The following datasets were generated:

| Author(s) | Year | Dataset title | Dataset URL | Database and Identifier |
|---|---|---|---|---|
| University of California Irvine | 2023 | Transcriptomes of whole blood of outbred Peromyscus leucopus and *Mus musculus* in response to lipopolysaccharide | https://www.ncbi.nlm.nih.gov/bioproject/?term=PRJNA975149 | NCBI BioProject, PRJNA975149 |
| University of California Irvine | 2022 | Peromyscus leucopus transcriptome in response to TLR agonists | https://www.ncbi.nlm.nih.gov/bioproject/?term=PRJNA874306 | NCBI BioProject, PRJNA874306 |
| University of California Irvine | 2023 | Transcriptomes of blood of Rattus norvegicus in response to lipopolysaccharide | https://www.ncbi.nlm.nih.gov/bioproject/?term=PRJNA973677 | NCBI BioProject, PRJNA973677 |
| University of California Irvine | 2023 | Transcriptome shotgun assembly of whole blood of female and male Peromyscus leucopus with and without treatment with lipopolysaccharide | https://www.ncbi.nlm.nih.gov/nuccore/GKOE00000000.1 | NCBI Nucleotide, GKOE00000000.1 |

The following previously published dataset was used:

| Author(s) | Year | Dataset title | Dataset URL | Database and Identifier |
|---|---|---|---|---|
| University of California Irvine | 2018 | Peromyscus leucopus transcriptome of Borrelia hermsii-infected and uninfected animals | https://www.ncbi.nlm.nih.gov/bioproject/?term=PRJNA508222 | NCBI BioProject, PRJNA508222 |

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
