## [Editor Report · eLife assessment]

This study provides a comprehensive whole genome transcriptomic analysis of three small mammals, including Peromyscus leucopus, after exposure to endotoxin lipopolysaccharide. The authors find that the inflammatory response of the three species is complex and that P. leucopus responds differently compared to mice and rats. The data are **convincing** and constitute an **important** advance in our understanding of inflammatory responses in animals that serve as reservoirs for relevant pathogens.

---

## [Referee Report · Reviewer #1 (Public Review)]

Summary:

o A well-executed series of experiments that will likely be of immense interest to (a) vector-borne disease researchers and (b) gram-negative sepsis/bacteremia researchers. The study uses comparative transcriptomics to begin probing what makes Peromyscus leucopus a unique host for numerous pathogens across the tree of life. Authors responded well to concerns raised in peer review and have produced an excellent second version of the manuscript.

Strengths:

o Use of outbred *M. musculus* is a commendable choice for the studies here.

o Use of both LPS and B. hermsii allows analysis of multiple different signaling pathways that may differ between the species.

o Upload of analyzed data onto Dryad is appreciated.

Weaknesses:

o None noted beyond the authors own limitation discussion section

---

## [Referee Report · Reviewer #2 (Public Review)]

Milovic, Duong, and Barbour investigate the inflammatory response of three species of small mammals (*P. leucopus*, *M. musculus*, and *R. norvegicus*) to endotoxin lipopolysaccharide (LPS) injection via genome-wide transcriptomics from blood samples. Understanding the inflammation response of *P. leucopus* is of importance as they are a reservoir for several pathogens. The study is a thorough, controlled, well researched analysis that will be valuable for designing and interpreting future studies. The authors discuss the limitations of the data and the potential directions. Clearly *P. leucopus* respond differently to the LPS exposure which is very interesting and opens the door for numerous other comparative studies.

The conclusions of the manuscript are thoughtful and supported by the data. The authors addressed my questions about mouse numbers, sex differences, and the presentation of Nos2 and Arg1 data.

---

## [Author Response]

The following is the authors’ response to the original reviews.

Re: Revised author response for eLife-RP-RA-2023-90135 (“The white-footed deermouse, an infection-tolerant reservoir for several zoonotic agents, tempers interferon responses to endotoxin in comparison to the mouse and rat” by Milovic, Duong, and Barbour”)

The revised manuscript has taken into account all the comments and questions of the two reviewers. Our responses to each of the comments are detailed below. In brief, the modifications or additional materials for the revision each specifically address a reviewer comment. These modifcations or materials include the following….

• a more in-depth consideration of sample sizes

• a better explanation of what p values signify for a GO term analysis

• a more detailed account of the selection of the normalization procedure for cross-species targeted RNA-seq (including a new supplemental figure)

• several more box plots in supplementary materials to complement the scatterplots and linear regressions of the figures of the primary text

• provision in a public access repository of the complete data for the RNA-seq analyses as well as primary data for figures and tables as new supplementary tables

• the expansion of description of the analysis done for the revision of Borrelia hermsii infection of P. leucopus. This included a new table (Table 10 of the revision) • development of the possible relevance of finding for longevity studies by citing similarities of the findings in P. leucopus with those in the naked mole-rat

• what we think is a better assessment of differences between female and male P. leucopus for this particular study, while still keeping focus on DEGs in common for females and males. This included a new figure (Figure 4 of the revision).

• removal of reference to a “inverse” relationship between Nos2 and Arg1 while still retaining ratios of informative value

We note that in the interval between uploading the original bioRxiv preprint and now we learned of the paper of Gozashti, Feschotte, and Hoekstra (reference 32), which supports our conception of the important place of endogenous retroviruses in the biology and ecology of deermice. This is the only addition or modification that was not a direct response to a reviewer comment or question, but it was germane to one of Reviewer #1’s comments (“Regarding..”).

**Reviewer #1:**
Supplemental Table 1 only lists genes that passed the authors statistical thresholds. The full list of genes detected in their analysis should be included with read counts, statistics, etc. as supplemental information.

We agree that provision of the entire lists of reference transcripts and the RNA-seq results for each of the 40 animals is merited. These datasets are too large for what the journal’s supplementary materials resource was intended for, so we have deposited them at the Dryad public access repository.

While P. leucopus is a critical reservoir for B. burgdorferi, caution should be taken in directly connecting the data presented here and the Lyme disease spirochete. While it's possible that P. leucopus have a universal mechanism for limiting inflammation in response to PAMPs, B. burgdorferi lack LPS and so it is also possible the mechanisms that enable LPS tolerance and B. burgdorferi tolerance may be highly divergent.

The impetus for the study was the phenomenon of tolerance of infection of P. leucopus by a number of different kinds of pathogens, not just B. burgdorferi. We take the reviewer’s point, though. Certainly, the white-footed deermouse is probably most notable at-large for its role as a reservoir for the Lyme disease agent. We doubt that the species responses to LPS and to the principal agonists of B. burgdorferi are “highly divergent”, though. Other than the TLR itself-TLR4 for LPS vs the heterodimer TLR2/TLR1 for the lipoproteins of these spirochetes--the downstream signaling is generally similar for amounts comparable in their agonist potency.

We had thought that we had addressed this distinction for B. burgdorferi and otherBorreliaceae members by referring to the earlier study. But we agree with the reviewer that what was provided on this point was insufficient in the context of the present work. Accordingly, for the revision we have added a new analysis of the data on experimental infection of P. leucopus with Borrelia hermsii, which lacks LPS and for which the TLR agonists eliciting inflammation are lipoproteins. We do this in a format (new Table 6) that aids comparison with the LPS experimental data elsewhere in the article. As the manuscript references, B. burgdorferi infection of P. leucopus elicits comparatively little inflammation in blood even at the height of infection. While this phenomenon with the Lyme disease agent was part of the rationale driving these studies, the better comparison with LPS was 5 days into B. hermsii infection when the animals are spirochetemic.

Statistical significance is binary and p-values should not be used as the primary comparator of groups (e.g. once a p-value crosses the deigned threshold for significance, the magnitude of that p-value no longer provides biological information). For instance, in comparing GO-terms, the reason for using of high p-value cutoffs ("None of these were up-regulated gene GO terms with p values < 1011 for *M. musculus*.") to compare species is unclear. If the authors wish to compare effect sizes, comparing enrichment between terms that pass a cutoff would likely be the better choice. Similarly, comparing DEG expression by p-value cutoff and effect size is more meaningful than analyses based on exclusively on p-value: "Of the top 100 DEGs for each species by ascending FDR p value." Description in later figures (e.g. Figure 4) is favored.

Effect sizes--in this case, fold-changes--were taken into account for GO term analysis and were specified in the settings that are described. So, any gene that was “counted” for consideration for a particular GO term would have passed that threshold and with a falsediscovery corrected p value of a specified minimum. There is no further scoring of the “hit” based upon the magnitude of the p value beyond that point. It is, as the reviewer writes, binary at that point. We are in agreement on those principles.

As we understand the comment above, though, the p-values referred to are in regard to the GO term analysis itself. The objective was discovery followed by inference. The situation was more like a genome-wide association study (GWAS) study. This is not strictly speaking a hypothesis test, because there was no stated hypothesis ahead of time or one driving the design. The “p value” for something like GO term analysis or GWAS provides an estimate of the strength of the association. It is not binary in that sense. The lower the p value, the greater confidence about the association. In a GWAS of a human population an association of a trait with a particular SNP or indel is usually not taken seriously unless the p value is less than 10^-7 or 10^-8. In the case of GO terms, the p value approximates (but is not equivalent to) the number of genes that are differentially expressed that belong to a GO cluster out of the total number of genes that define that cluster. The higher the proportion of the genes in the cluster that are associated with a treatment (LPS vs. saline), the lower the p value. Thus, it provides information beyond the point at which it would be rightly deemed of little additional value in many hypothesis testing circumstances.

That said, we agree that the original manuscript could have been clearer on this point and have for the revision expanded the description of the GO term analysis in the Methods, including some explanation for a reader on what the p value signifies here. We also refrain from specifying a certain p value for special attention and merely list 20 by ascending p value.

The ability to use of CD45 to normalize data is unclear. Authors should elaborate both on the use of the method and provide some data how the data change when they are normalized. For instance, do correlations between untreated Mus and Peromyscus gene expression improve? The authors seem to imply this should be a standard for interspecies comparison and so it would be helpful to either provide data to support that or, if applicable, use of the technique in literature should be referenced.

The reviewer brings up an important point that we considered addressing in more depth for the original manuscript but in the end deferred to considerations about length and left it out.

But we are glad to address this here, as well as in the revised manuscript.

We did not intend to imply either that this particular normalization approach had been done before by others or that it “should” be a standard. We are not aware of another report on this, and it would be up to others whether it would be useful or not for them. We made no claim about its utility in another model or circumstance. The challenge before us was to do a comparative analysis of transcription in the blood not just for animals of one species under different conditions but animals of two different genera under different conditions. A notable difference between the animals was in their white blood cell counts, as this study documents. White cells would be the source of a majority of transcripts of potential relevance here, but there would also be mRNA for globins, from reticulocytes, from megakaryocytes, and likely cell-free RNA with origins in various tissues. If the white cell numbers differed, but the non-white cell sources of RNA did not, then there could be unacknowledged biases.

It would be like comparing two different kinds of tissues and assuming them to be the same in the types and numbers of cells they contained. Four hours after a dose of LPS the liver cells (or brain cells) would differ in their transcriptional profiles from untreated the livers (or brains) of untreated animals for sure, but there would not be much if any change in the numbers of different kinds of cells in the liver (or brain) within 4 hours. The blood can change a lot in composition within that time frame under these same conditions. Some sort of accounting for differing white cell numbers in the blood in different outbred animals of two species seemed to be called for.

The normalization that was done for the genome-wide analysis was not based on a particular transcript, but instead was based on the total number of reads, the lengths of the reference transcripts, and the distributions of reads matching to the tens of thousands of references for each sample. This was done according to what are standard procedures by now for bulk RNAseq analyses. Because the reference transcript sets for *P. leucopus* and *M. musculus* differed in their numbers and completeness of annotation, we did not attempt any cross-species comparison for the same set of genes at that point. That would not be possible because they were not entirely commensurate.

The GO term analysis of those results provided the leads for the more targeted approach, which was roughly analogous to RT-qPCR. For a targeted assay of this sort, it is common to have a “housekeeping gene” or some other presumably stably transcribed gene for normalization. A commonly used one is Gapdh, but we had previously found that Gapdh was a DEG itself in the blood in *P. leucopus* and *M. musculus* at the four hour mark after LPS. The aim was to provide for some adjustment so datasets for blood samples differing in white blood cell counts could be compared. Two options were the 12S ribosomal RNA of the mitochondria, which would be in white cells but not mature erythrocytes, and CD45, which has served an approximately similar function for flow cytometry of the blood. As described in what has been added for the revision and the supplementary materials, we compared these different approaches to normalization. Ptprc and 12S rRNA were effectively interchangeable as the denominator with identifying DEGs of *P. leucopus* and *M. musculus* and cross-species comparisons.

Regarding the ISG data-is a possible conclusion not that Peromyscus don't upregulate the antiviral response because it's already so high in untreated rodents? It seems untreated Peromyscus have ISG expression roughly equivalent to the LPS mice for some of the genes. This could be compared more clearly if genes were displayed as bar plots/box and whisker plots rather than in scatter plots. It is unclear why the linear regression is the key point here rather than normalized differences in expression.

In answer to the question: yes, that is possible. In the interval between uploading of the manuscript and this revision, we became aware of a study by Gozashti and Hoekstra published this year in Molecular Biology and Evolution (reference 32) and reporting on the “massive invasion” of endogenous retroviruses in P. maniculatus and the defenses deployed in response to achieve silencing. We cite this work and discuss it, including related findings for P. leucopus, in the revision.

We had originally intended to include box plots as well as scatterplots with regressions for the data, but thought it would be too much and possibly considered redundant. But with this encouragement from the reviewer we provide additional box plots in supplementary materials for the revision.

Some sections of the discussion are under supported:The claim that low inflammation contributes to increased lifespan is stated both in the introduction and discussion. Is there justification to support this? Do aged pathogen-free mice show more inflammation than aged Peromyscus?

We respectively point out that there was not a claim of this sort. We stated a fact about P. leucopus’ longevity. We made no statement connecting longevity and inflammation beyond the suggestion in the introduction that the explanation(s) for infection tolerance might have some bearing for studies on determinants of life span.

But the reviewer’s comment prompted further consideration of this aspect of Peromyscus biology. This led eventually to the literature on the naked mole-rat, which seems to be the rodent with the longest known life span and the subject of considerable study. The discussion section of the revision has an added paragraph on some of the similarities of P. leucopus and the naked mole-rat in terms of neutrophils, expression of nitric oxide synthase 2 in response to LPS, and type 1 interferon responses. While this is far from decisive, it does serve to connect some of the dots and, hopefully, is considered at least partially responsive to the reviewer’s question.

The claim that reduced Peromyscus responsiveness could lead to increased susceptibility to infection is prominently proposed but not supported by any of the literature cited.

There was not this claim. In fact, it was framed as a question, not a statement. Nevertheless, we think we understand what the comment is getting at and acknowledge in the revision that there may be unexamined circumstances in which P. leucopus may be more vulnerable.

References to B. burgdorferi, which do not have LPS, in the discussion need to ensure that the reader understands this and the potential that responses could be very different.

We think we addressed this comment in a response above.

**Reviewer #2:**
1. How were the number of animals for each experiment selected? Was a power analysis conducted?

A power analysis of any meaning for bulk RNA-seq with tens of thousands of reference transcripts, each with their own variance, and a comparison of animals of two different genera is not straight forward. Furthermore, a specific hypothesis was not being tested. This was a broad, forward screen. But the question about sample sizes is one that deserves more attention than the original manuscript provided. This now provided in added text in two places in Methods ( “RNA-seq” and “Genome-wide different gene expression”) in the revision.

1. The authors conducted a cursory evaluation of sex differences of P. leucopus and reported no difference in response except for Il6 and Il10 expression being higher in the males than the females in the exposed group. The data was not presented in the manuscript. Nor was sex considered for the other two species. A further discussion of the role that sex could play and future studies would be appreciated.

We agree that the limited analysis of sex differences and the undocumented remark about Il6 and Il10 expression in females and males warranted correction. For the revision we removed that analysis of targeted RNA-seq of P. leucopus from the two different studies. For this study we were looking for differences that applied to both species. This was the reason that there were equal numbers of females and males in the samples. We agree that further investigation of differences between sexes in their responses is of interest but is probably best left for “future studies”.

But in revision we do not entirely ignore the question of sex of the animal and provide an additional analysis of the bulk RNA-seq for P. leucopus with regard to differences between females and males. This basically demonstarted an overall commensurability between sexes, at least for the purposes of the GO term analysis and subsequent targeted RNA-seq, but did reveal some exceptions that are candidate genes for those future studies.

In the revision, we also add for the discussion and its “study limitations” section a disclaimer about possibly missing sex associated differences because the groups were mixed sexes.

1. The ratio of Nos2 and Arg1 copies for LPS treated and control *P. leucopus* and *M. musculus* in Table 3 show that in *P. leucopus* there is not a significant difference but in *M. musculus* there is an increase in Nos2 copies with LPS treatment. The authors then used a targeted RNA-seq analysis to show that in *P. leucopus* the number of Arg1 reads after LPS treatment is significantly higher than the controls. These results are over oversimplified in the text as an inverse relationship for Nos2/Arg1 in the two species.

We agree. In addition to providing box plots for Arg1 and Nos2, as suggested by Reviewer #1, we also replaced “ratio” in commenting on Arg1 and Nos2, with “differences in Nos2 and Arg1 expresssion” replacing “ratio of Nos2 to Arg1 expression” at one place. At another place we have removed “inverse” with regard to Nos2 and Arg1. But we respectfully decline to remove Nos2/Arg1 from Figure 5 (now Figure 6) or inclusion of Nos2/Arg1 ratios elsewhere. According to our understanding there need not be an inverse relationship for a ratio to have informative value.

**Recommendations For the Authors**

We thank the two reviewers for their constructive recommendations and suggestions, in some case pointing out errors we totally missed. For the great majority, the recommendations were followed. Where we decline or disagree we explain this in the response.

**Reviewer #1 (Recommendations For The Authors):**
• How was the FDR < 0.003 cutoff chosen for DEG? All cutoffs are arbitrary but there should be some justification.

We agree and have provided the rationale at that point in the paper (before Figure 3) in R2:"For GO term analysis the absolute fold-change criterion was ≥ 2. Because of the ~3-fold greater number of transcripts for the *M. musculus* reference set than the *P. leucopus* reference set, application of the same false-discovery rate (FDR) threshold for both datasets would favor the labeling of transcripts as DEGs in *P. leucopus*. Accordingly, the FDR p values were arbitrarily set at <5 x 10-5 for P. leucopus and <3 x 10-3 for *M. musculus* to provide approximately the same number of DEGs for *P. leucopus* (1154 DEGs) and *M. musculus* (1266 DEGs) for the GO term comparison."

• It would be helpful to include a figure demonstrating the correlation between CD45 and WBC ("Pearson's continuous and Spearman's ranked correlations between log-transformed total white blood cell counts and normalized reads for Ptprc across 40 animals representing both species, sexes, and treatments were 0.40 (p = 0.01) and 0.34 (p = 0.03), respectively.")

In both the first version of the revision (R1) and in R2 we provide a fuller explanation of the choice of CD45 (Ptprc) for normalization as detailed in the response to Reviewer #1's public comment. In the revision only Pearson's correlation and p value is given. We did not think another figure was justified after there was additional space devoted to this in both R1 and R2.

• Unclear what the following paragraph is referring to-is this from the previous paper? Was this experiment introduced somewhere? "Low transcription of Nos2 and high transcription of Arg1 both in controls and LPS-treated P. leucopus was also observed in the experiment where the dose of LPS was 1 µg/g body mass instead of 10 µg/g and the interval between injection and assessment was 12 h instead of 4 h (Table 4)."

This experiment is described in the Methods in the original and subsequent versions, but we agree that it is not clear whether it was from present study or previous one. Here is the revised text for R2:"Low transcription of Nos2 in both in controls and LPS-treated P. leucopus and an increase inArg1 with LPS was also observed in another experiment for the present study where the dose of LPS was 1 µg/g body mass instead of 10 µg/g and the interval between injection and assessment was 12 h instead of 4 h (Table 4)."

• Regarding the differences in IFNy between outbred and BALB/c mice-are there any other RNA-seq datasets you can mine where other inbred mice (B/6, C3H, etc) have been injected with LPS and probed roughly the same amount of time later? Do they look like BALB/c or the outbreds?

In both the original and R1 and R2 we cite two papers on the difference of BALB/c mice. While this is of interest for follow-up in the future, we did not think additional content on a subject that mainly pertains to *M. musculus* was warranted here, where the main focus is Peromyscus.

• Figure 8 and its legend are difficult to follow. The top half of the figure is not well explained and it's unclear what species this is. Decreased use of abbreviations would help. Consider marking each R2 value as Mus or Peromyscus (As done in Fig 9). There are some typographical errors in the legend ("gree,") incomplete sentence missing the words LPS or treatment AND Mus: "Co-variation between transcripts for selected PRRs (yellow) and ISGs (gree) in the blood of P. leucopus (P) or (M) with (L") or without (C)."

This is now Figure 9 in both R1 and R2. We revised it for R1 to include references to the box plots in supplementary materials, but agree with Reviewer #1's recommendation to correct the typos and make the legend less confusing. We did not think that further labeling of the R2 values in the scatterplots with the species names was necessary. The data points are not just colors but also different symbols, so it should be fairly easy for readers to distinguish the regression lines by species. For R2 this is the revised legend with additions in response to the recommendation underlined:

"Figure 9. Co-variation between transcripts for selected PRRs and ISGs in the blood of *P. leucopus* (P) or *M. musculus* (M) with (L) or without (C) LPS treatment. Top panel: matrix of coefficients of determination (R2) for combined *P. leucopus* and *M. musculus* data. PRRs are indicated by yellow fill and ISGs by blue fill on horizontal and vertical axes. Shades of green of the matrix cells correspond to R2 values, where cells with values less than 0.30 have white fill and those of 0.90-1.00 have deepest green fill. Bottom panels: scatter plots of log-transformed normalized Mx2 transcripts on Rigi (left), Ifih1 (center), and Gbp4 (right). The linear regression curves are for each species. For the right-lower graph the result from the General Linear Model (GLM) estimate is also given. Values for analysis are in Table S4; box plots for Gbp4, Irf7, Isg15,Mx2, and Oas1 are provided in Figure S6."

• Discussion section could benefit from editing for clarity. Examples listed: o Unclear what effect is described here "The bacterial infection experiment indicated that the observed effect in P. leucopus was not limited to a TLR4 agonist; the lipoproteins of B. hermsii are agonists for TLR2 (Salazar et al. 2009)."

Both R1 and R2 include the new section on the B. hermsii infection model. This was added in response to Reviewer #1 public comment. So the expanded consideration of this aspect should address the reviewer's recommendation for more clarity and context here. For R2 we modified the text in the discussion of R1:

"The analysis here of the B. hermsii infection experiment also indicated that the phenomenon observed in P. leucopus was not limited to a TLR4 agonist."

o Unclear what the takeaway from this paragraph is: "Reducing the differences between *P. leucopus* and the murids *M. musculus* and *R. norvegicus* to a single all-embracing attribute may be fruitless. But from a perspective that also takes in the 2-3x longer life span of the whitefooted deer mouse compared to the house mouse and the capacity of P. leucopus to serve as disease agent reservoir while maintaining if not increasing its distribution (Moscarella et al. 2019), the feature that seems to best distinguish the deer mouse from either the mouse or rat is its predominantly anti-inflammatory quality. The presentation of this trait likely has a complex, polygenic basis, with environmental (including microbiota) and epigenetic influences. An individual's placement is on a spectrum or, more likely, a landscape rather than in one or another binary or Mendelian category."

We agree that modification, simplication, and clarification was called for. In response to a public comment of Reviewer #1 we had changed that section, leaving out reference to longevity here.Here is the revised text in both R1 and R2:

"Reducing differences between P. leucopus and murids *M. musculus* and *R. norvegicus* to a single attribute, such as the documented inactivation of the Fcgr1 gene in P. leucopus (7), may be fruitless. But the feature that may best distinguish the deermouse from the mouse and rat is its predominantly anti-inflammatory quality. This characteristic likely has a complex, polygenic basis, with environmental (including microbiota) and epigenetic influences. An individual’s placement is on a spectrum or, more likely, a landscape rather than in one or another binary or Mendelian category."

Minor comments:• Use of blue and red in figures as the -only- way to easily distinguish between groups is a poor choice-both in terms of how inclusivity of color-blind researchers and enabling grayscale printing. Most detrimental in Figure 2, but also slightly problematic in Figure 1. Use of color and shape (as done in other figures) is a much better alternative.

We agree. Both figures have been modified to include an additional characteristic for denoting the data point. For Figure 1 it is a black filling, and for Figure 2 it is the size of symbol in additon to the color. This should enable accurate visualization by color blind individuals and printing in gray scale. We have added definitions for the symbols within the graph itself, so there is no need to refer to the legend to interpret what they mean.

• Note the typo where it should read P leucopus: "The differences between *P. musculus* and *M. musculus* in the ratios of Nos2/Arg1 and IL12/IL10 were reported before (BalderramaGutierrez et al. 2021),"

We thank the reviewer for pointing this typo out, which also carried over to R1. It has been corrected for R2.

• Optional: Can the relationship between the ratios in figure 5 and macrophage "types" be displayed graphically alongside the graphs? It's a little challenging to go back and forth between the text and the figure to try to understand the biological implication.

We considered something like this but in the end decided that we were not yet comfortable assigning “types” in this fashion for Peromyscus.

**Reviewer #2 (Recommendations For The Authors):**
• Be consistent with nomenclature for your species/treatment groups in the text, figures, and tables. For example, you go back and forth between "P. leucopus" and "deermouse" in the text.And in figures you use "P," "Peromyscus", or "Pero".

In the Methods section of the original and revisions R1 and R2 we indicate that "deermouse" is synonymous with "Peromyscus leucopus" and "mouse" is synonymous with "*Mus musculus*" in the context of this paper. We think that some alternation in the terms relieves the text of some of its repetitiveness and that readers should not have a problem with equating one with the other. The use of "deermouse" also reinforces for readers that Peromyscus is not a mouse. With regard to the abbreviations for P. leucopus, those were used to accommodate design and space issues of the figures or tables. In all cases, the abbreviations referred to are defined in the legends of the figures. So, we respectfully decline to follow this recommendation.

• Often the sentence structure and/or word choice is irregular and makes quick/easy comprehension difficult. Several examples are:o The third paragraph of the introduction

We agree that the first and second sentences are unclear. Here is the revision for R2:

“As a species native to North America, P. leucopus is an advantageous alternative to the Eurasian-origin house mouse for study of natural variation in populations that are readily accessible (9, 53). A disadvantage for the study of any Peromyscus species is the limited reagents and genetic tools of the sorts that are applied for mouse studies.”

o The first line after Figure 5 on page 9.

We agree. The long sentence which we think the reviewer is referring to has been in split into two sentences for R2.

“An ortholog of Ly6C (13), a protein used for typing mouse monocytes and other white cells, has not been identified in Peromyscus or other Cricetidae family members. Therefore, for this study the comparison with Cd14 is with Cd16 or Fcgr3, which deermice and other cricetines do have.”

o The sentence that starts "Our attention was drawn to..." on page 14.

We agree that the sentence was awkward and split into two sentences.

“Our attention was drawn to ERVs by finding in the genome-wide RNA-seq of LPS-treated and control rats. Two of the three highest scoring DEGs by FDR p value and fold-change were a gagpol polyprotein of a leukemia virus with 131x fold-change from controls and a mouse leukmia virus (MLV) envelope (Env) protein with 62x fold-change (Dryad Table D5).”

• For figures with multiple panels, use (A, B) etc then indicate which panel you are discussing in your text. This is a very data heavy study and your readers can easily get lost.

We agree and have added pointers in the text to the panels we are referring to. But we prefer to use easily understood descriptors like “left” and “upper” over assigned letters.

• For all the figures, where are the stats from the t-tests? Why didn't you do a two-way ANOVA? Instead of multiple t-tests?

Where we are not hypothesis testing and we are able to show all the data points in box-whisker plots with distributions fully revealed, our default position is not to apply significance tests in a post hoc fashion. If a reader or other investigator wants to do this for other purposes, e.g. a meta-analysis, the data is provided in public repository for them to do this. We are not sure what the reviewer means by "multiple t-tests" for "all figures". Where we do 2-tailed t-tests for presentation of data for many genes in a table for the targeted RNA (where individual values cannot shown in the table), there is always correction for multiple testing, as indicated in Methods. The p values shown as "FDR" are after correction.

• Results paragraph "LPS experiment and hematology studies"o List the two species for the first description to orient the reader since you eventually include rat data.

We agree that this is warranted and followed this recommendation for R2.

o Not all the mice experienced tachypnea, but the text makes it seem like 100% did.

We are not sure what the reviewer is referring to here. This is what is in the text on tachypnea: "By the experiment’s termination at 4 h, 8 of 10 *M. musculus* treated with LPS had tachypnea, while only one of ten LPS-treated P. leucopus displayed this sign of the sepsis state (p = 0.005)." The only other mention of "tachypnea" was in Methods.

• Figure 1: Why was the *M. musculus* outlier excluded? Where any other outliers excluded?

That data point for the mouse was not "excluded" from the graph. It is identified (MM17) for reference with Table 1, and there is the graph for all to see where it is. It was only excluded from the regression curve for control mice. There was no significance testing. There were no other outliers excluded.

• Figure 3: explain the colors and make the scales the same for all the panels or at least for the upregulated DEGs and the downregulated DEGs.

We have modified the legend for Figure 3 to include fuller definitions of the x-axes and a description of the color spectrum. We decline to make the x-axis scale the same for all the panels because the horizontal bars in “transcription down” panels would take up only a small fraction of the space. The x-axes are clearly defined and the colors of the bars also indicate the differences in p-values. We doubt that readers will be misled. Here is the revised legend: “Figure 3. Gene Ontology (GO) term clusters associated with up-regulated genes (upper panels) and down-regulated genes (lower panels) of *P. leucopus* (left panels) and *M. musculus* (right panels) treated with LPS in comparison with untreated controls of each species. The scale for the x-axes for the panels was determined by the highest -log10 p values in each of the 4 sets. The horizontal bar color, which ranges from white to dark brown through shades of yellow through orange in between, is a schematic representation of the -log10 p values.”

• Results paragraph "Targeted RNA seq analysis"o In the third paragraph, an R2 of 0.75 is not close enough to 1 to call it "~1"

What the reviewer is referring to is no longer in either R1 and R2, as detailed in the authors' response to public comments.

o In the 4th paragraph, where are your stats?

We have replaced terms like "substantially" and "marginally" with simple descriptions of relationships in the graphs.

"For the LPS-treated animals there was, as expected for this selected set, higher expression of the majority genes and greater heterogeneity among *P. leucopus* and *M. musculus* animals in their responses for represented genes. In contrast to the findings with controls, Ifng and Nos2 had higher transcription in treated mice. In deermice the magnitude of difference in the transcription between controls and LPS-treated was less."

• Figure 4: The colors are hard to see, I suggest making all the up regulated reads one color, the down regulated reads a different color, and the reads that aren't different black or gray.

This is now Figure 5 in R1 and R2. The selected genes that are highlighted in the panels are denoted not only by color but also by type of symbol. We do not think that readers will have a problem telling one from another even if color blind. The purpose of this figure was to provide an overview and a visual representation with calling out of selected genes, some of which will be evaluated in more detail later. We thought that this was necessary before diving deeper into the data of Table 2. We do not think further discriminating between transcripts in the categorical way that the reviewer suggests is warranted at this point. So, we respectfully decline to follow this suggestion.

• Results paragraph " Alternatively- activated macrophages...."o Include a brief description of Nos2 and Arg1

We have defined what enzymes these are genes for in R2.

o How do you explain the lack of a difference in P. leucopus Arg1? Your text says the RT-qPCR confirms the RNA-seq findings.

There was a difference in *P. leucopus* Arg1 by RT-qPCR between control and LPS treated by about 3-fold. By both RNA-seq and RT-qPCR Arg1 transcription is higher in *P. leucopus* than in *M. musculus* under both conditions. But we have modified the sentence so that does not imply more than what the data and analysis of the table reveal.

"While we could not type single cells using protein markers, we could assess relative transcription of established indicators of different white cell subpopulations in whole blood. The present study, which incorporated outbred *M. musculus* instead of an inbred strain, confirmed the previous finding of differences in Nos2 and Arg1 expression between *M. musculus* and *P. leucopus* (Figure 5; Table 2). Results similar to the RNA-seq findings were obtained with specific RT-qPCR assays for Nos2 and Arg1 transcripts for *P. musculus* and *M. musculus* (Table 3)."

• Figure 5: reorganize the panels to make the text description and label with letters, where are the stats?

We thought the figure (now Figure 6) was self-explanatory, but agree that further explanation in the legend was indicated. We prefer to use descriptions of locations (“upper left”) over labels, like “panel C”, which do not obviously indicate the location of the panel. Of course, if the journal’s style mandates the other format we will do so. Our response about “stats” for boxplot figures is the same as what we provided above.

• Results paragraph "Interferon-gamma and interleukin-1 beta..."o Either add the numbers or direct the viewer to where Ifng is in Table 2. The table is very big and Ifng is all the way at the bottom!

We agree that this table is large, but we thought it better to err on the side of inclusiveness by having a single table, rather than have some genes in the main article and other results in a supplementary table. We thought that it would make it easier for reviewers and readers to find a gene of interest, but we also acknowledge the challenge to locate the genes we highlight. We follow for R2 that reviewer's recommendation to provide some guidance for readers trying to locate a featured gene by pointing relative locations. While adding a column of numbers to already complex table seems more than what is called for, we are depositing an Excel spreadsheet of the table at the Dryad repository to facilitate searching by an interested reader for a particular gene.

• Figure 6: stats? The pink and red are hard to easily distinguish from each other. I also suggest not using red and green together for color blind readers.

With regard to the box-plots and significance testing, please see response above to an earlier recommendation. We have removed an interpretative adjective (i.e. "marked") from the description of the graph. Different symbols as well as colors are used, so we do not think that this will pose a problem for readers, even those with complete red-green color blindness. For what it’s worth, with regard to the "red" and "pink" issue, according to the figure on our displays the colors of the two symbols appear to be red and purple. They are also applied to different species and different conditions for those species.

• Figure 8: In the legend it says "... PRRs (yellow) and ISGs (gree)" which is a typo, but don't you mean blue not green anyways?

See response above to Reviewer #1's recommendation. This has been corrected.